nuclear chemistry/volcanology/geochemistry

radon emissions, subvolcanic thermal gradients, temperature-dependent radon diffusion

**Author for correspondence:**
Silvio Mollo
e-mail: silvio.mollo@uniroma1.it

# Transient to stationary radon ($^{220}$Rn) emissions from a phonolitic rock exposed to subvolcanic temperatures

Silvio Mollo[1,2], Paola Tuccimei[3], Michele Soligo[3], Gianfranco Galli[2], Gianluca Iezzi[2,4] and Piergiorgio Scarlato[2]

[1]Dipartimento di Scienze della Terra, Sapienza Università di Roma, P.le Aldo Moro 5, 00185 Roma, Italy
[2]Istituto Nazionale di Geofisica e Vulcanologia, Via di Vigna Murata 60, 00143 Roma, Italy
[3]Dipartimento di Scienze, Università 'Roma Tre', Largo S. L. Murialdo 1, 00146 Roma, Italy
[4]Dipartimento di Ingegneria & Geologia, Università G. d'Annunzio, Via dei Vestini 30, 66013 Chieti, Italy

 SM, 0000-0002-1448-0282

Rock substrates beneath active volcanoes are frequently subjected to temperature changes caused by the input of new magma from the depth and/or the intrusion of magma bodies of variable thickness within the subvolcanic rocks. The primary effect of the influx of hot magma is the heating of surrounding host rocks with the consequent modification of their physical and chemical properties. To assess mobilization in subvolcanic thermal regimes, we have performed radon ($^{220}$Rn) thermal experiments on a phonolitic lava exposed to temperatures in the range of 100–900°C. Results from these experiments indicate that transient Rn signals are not unequivocally related to substrate deformation caused by tectonic stresses, but rather to the temperature-dependent diffusion of radionuclides through the structural discontinuities of rocks which serve as preferential pathways for gas release. Intense heating/cooling cycles are accompanied by rapid expansion and contraction of minerals. Rapid thermal cycling produced both inter- and intra-crystal microfracturing, as well as the formation of macroscopic faults. The increased number of diffusion paths dramatically intensified Rn migration, leading to much higher emissions than temperature-dependent transient changes. This geochemical behaviour is analogous to positive anomalies recorded on active volcanoes where dyke injections produce thermal stress and deformation in the host rocks. An increased Rn signal far away from the location of a

magmatic intrusion is also consistent with microfracturing of subsurface rocks over long distances via thermal stress propagation and the opening of new pathways.

# 1. Introduction

There is growing recognition that radon monitoring represents an important study area for the investigation of precursory signals in active volcanic settings and for interpreting geochemical anomalies before volcanic eruptions (e.g. [1–8]). Volcanic activity is commonly preceded over days to months by increased volcanic tremors and marked increases in Rn (i.e. $^{222}$Rn and $^{220}$Rn) emissions due to magmatic intrusions and deformation within the flanks of the volcanic edifice [9,10]. Small dyke injections may generate localized stress along faults and fissures in the volcanic pile, causing marked Rn anomalies [11]. Conversely, large volumes of intrusive magma cause substantial changes in the stress field within the subvolcanic rock substrate, so that Rn anomalies can be measured over distances of several tens to hundreds of kilometres [12–14]. Thermal effects due to heat flow and increasing volcanic temperatures for months or even years are frequently accompanied by high Rn emissions [15,16]. Intense heating facilitates the extraction of Rn from subvolcanic rocks, and sometimes the increase in Rn emissions scales with the rate of magma ascent and the release of other volcanic gases through the fractured rocks [17]. Temperature gradients and carrier gases may also induce transient signals where Rn emissions are spatially heterogeneous and temporarily variable [11,18,19].

Subvolcanic rock substrates beneath active volcanoes are highly dynamic environments in which the geological materials are subjected to substantial physical and chemical changes caused by the injection of magmas from depth [20–25]. The ascending magma batches may stall at very shallow levels, even within the lava pile of the volcanic edifice, or may feed complex dyke networks sometimes intruding only a few metres beneath the ground surface. Heat flowing from hotter magma bodies into colder host rocks of the subvolcanic substrate [26] may cause cyclic heating and cooling stages [27]. Heat conduction within rocks of variable thickness produces temperature differences of the order of approximately 100–1000°C. These thermal gradients occur over distances of thousands of metres, incorporating several cubic kilometres of subvolcanic rock [28,29]. Thermal stresses within the crystalline lithologies produce expansion and contraction of the constituent minerals, frequently forming a network of pervasive microfractures [30,31]. Despite a gamut of experimental studies investigating the effect of rock deformation on Rn emission [32–42], the role played by subvolcanic thermal regimes on the Rn fluxes from subvolcanic rocks and the background level of Rn emissions has received much less attention [29,43]. In order to address this paucity of information, we have conducted Rn ($^{220}$Rn) thermal experiments on a phonolitic lava exposed to increasing temperatures and variable heating/cooling cycles. Results from real-time Rn monitoring across a thermal range of 100–900°C allows for a better understanding and quantification of the relationship between Rn signal and magmatic activity during thermally induced physical and chemical changes within subvolcanic rocks.

# 2. Methods

## 2.1. Starting material

A massive phonolite from the Colli Albani volcanic district (Latium, Italy) was used for the Rn thermal experiments (cf. [39]). The Colli Albani volcanic district is near to the city of Rome and its period of activity, from 608 to 36 ka, was characterized by effusive and explosive eruptions fed by silica-undersaturated ultrapotassic magmas [44–46]. The rock selected for Rn thermal experiments is a phonolite composed of a millimetre- to submillimetre-sized phenocrysts of leucite, clinopyroxene (augite) and magnetite, in order of abundance, and submillimetre-sized matrix minerals that are identical to the phenocryst assemblage. The petrophysical properties of the phonolite have already been investigated by Mollo *et al.* [39]: the deformation behaviour is characteristically brittle due to microfracturing, the total rock porosity ($\phi$) is 3.6%, and P-wave ($V_p$) and S-wave ($V_s$) velocities are 5.51 and 3.84 km s$^{-1}$, respectively. For the purpose of this study, the same block of phonolite was cored to obtain cylindrical rock samples 50 mm in diameter and 110 mm in length.

## 2.2. Radon thermal experiments

The experimental set-up used to perform the Rn thermal experiments consists of a furnace equipped with a Rn monitoring system specifically designed and developed at the HP-HT Laboratory of Experimental Volcanology and Geophysics of the Istituto Nazionale di Geofisica e Vulcanologia (INGV) in Rome (Italy), in order to analyse the Rn signal emitted from rocks exposed to subvolcanic temperatures. A detailed description of the experimental set-up is reported in [29] and is briefly summarized in the next section.

This work aims to better delineate the effects of background thermal gradients typical of active volcanic areas on the Rn emissions from crystalline rocks that frequently compose subvolcanic basements. Time- and temperature-dependent variations in the Rn signal have been investigated under the following thermal conditions:

— EXP1. The rock sample was first kept at 100°C for 48 h. Then, through a heating ramp ($\Delta t$) of 5 min, the temperature was increased ($\Delta T$) by 100°C and maintained constant for 22 h. This incremental step was replicated eight times (5 min $\Delta t$ and 100°C $\Delta T$) up to the maximum experimental temperature of 900°C.
— EXP2. The rock sample was first kept at 300°C for 48 h and, then, was subjected to low- and high-thermal stress cycles. For the low-temperature stress cycles (300°C $\Delta T$), the rock sample was heated from 300 to 600°C (5 min $\Delta t$) and, after 1 h, was cooled down to 300°C at the same rate. Note that the dwell time of 1 h was sufficient to analyse the Rn signal [38] and to ensure thermal homogenization within the rock sample [47]. At the end of the third cycle, the temperature was maintained constant at 300°C for 48 h. The same strategy was applied for the high-temperature thermal stress cycles (600°C $\Delta T$) by increasing the temperature from 300 to 900°C.
— EXP3. More intense heating/cooling cycles were performed with the aim to thermally stress the rock sample and to induce microfracturing. The rock sample was heated from 100 to 900°C (5 min $\Delta t$) and was then cooled from 900 to 100°C at the same rate. This heating/cooling cycle (5 min $\Delta t$ and 800°C $\Delta T$) was repeated 20 times. Owing to the short acquisition time, the Rn signal was not monitored during the very fast heating/cooling steps. However, at the end of the 20th cycle, the temperature of the furnace was decreased down to 300°C and maintained constant for 120 h, in order to measure the Rn change.
— EXP4. The thermally stressed rock sample obtained from EXP3 was disaggregated along the main macroscopic fault planes caused by the extremely fast heating/cooling cycles. Seven large rock fragments with centimetre dimensions were recovered. All the rock fragments were placed in the furnace preheated at 300°C, and the Rn signal monitored for 120 h.

EXP1 was designed to reproduce the temperature-dependent molecular transport of Rn from phonolitic rocks exposed to prograde thermal conditions, such as during a single injection of a dyke into subvolcanic rocks. By contrast, EXP2, EXP3 and EXP4 were performed to investigate the response of Rn emission in a highly dynamic regime characterized by a complex network of dykes causing cyclic heating and cooling of the subvolcanic rocks. To better constrain the effects of variable degrees of thermal cycling, the magnitude of heating/cooling cycles adopted for EXP2 were significantly less than those used for EXP3 and EXP4.

## 2.3. Radon measurement and correction method

Rn gas emitted from the rock samples was measured through an Rn monitor (RAD7, Durridge Company Inc., USA) connected to the furnace in a closed-loop configuration and equipped with a solid-state silicon alpha detector [29]. The most important characteristics of this novel set-up can be summarized as follows:

— the cylindrical rock sample is suspended in a gas-impermeable alumina tube of the vertical furnace;
— the rock sample is located in the isothermal hot zone of the furnace and the temperature is monitored on the rock surface by a Type S thermocouple with an uncertainty of ±3°C;
— gas leakage is prevented by upper and lower gas-tight flanges sealing the alumina tube;
— a Teflon tube inserted into the upper flange connects, in a closed-loop configuration, the alumina tube to a gas-drying unit filled with a desiccant ($CaSO_4$) to the RAD7;
— a recirculating pump moves the gas from the alumina tube towards the RAD7 solid-state detector for alpha counting;

— the gas is recirculated continuously through the system and ambient air is excluded from the experimental set-up during the run;

— the electrostatic detector collects the charged ions and discriminates the electrical pulses generated by the alpha particles of [216]Po for rapid determination of [220]Rn;

— the activity concentration of [220]Rn is measured continuously during experiments with an acquisition time (i.e. cumulative time) of 1 h for each single measurement for which the uncertainty of the mean at 95% confidence level is then calculated (electronic supplementary material table S1);

— according to the statistical analysis reported in Tuccimei *et al.* [38], the experimental uncertainty decreases significantly with increasing temperature due to the higher velocity of gas particles. With respect to room temperature, there is a significant improvement in data resolution when $T$ approaches approximately 100°C and the uncertainty of the [220]Rn mean at 95% confidence interval decreases from 5% to 2.7%.

It is worth stressing that Rn is released from the crystal lattice into the adjacent pore space and fracture network via alpha recoil. Owing to the short recoil length ($3 \times 10^{-8}$ cm), only atoms produced at the crystal surface are released from the rock to the surrounding medium. This first step is probably dominated by alpha recoil, which is temperature independent and characterized by the poor intrinsic mobility of Rn atoms [19–24,26–38]. In the second step, Rn moves through the pore and fracture network. If there is no pressure gradient, this process is dominated by diffusion which is highly temperature dependent [19–24,26–38]. In this study, we focus on this aspect by investigating the molecular diffusive transport of [220]Rn. Laboratory investigations are greatly facilitated by the short half-life of [220]Rn (56 s) compared with that of [222]Rn (3.82 days) [29,37–39]. The [220]Rn atoms rapidly reach the equilibrium activity concentration and respond almost instantaneously to any physical and chemical change of the rock samples. The geochemical behaviour of [220]Rn is identical to that of [222]Rn, due to the lack of isotopic fractionation between both isotopes with an extremely low mass difference of 0.01% [38].

To correctly interpret the experimental data, a decay ($D$) correction was applied to the measured signal ([220]Rn$_M$), by considering the travel time of atoms during gas transport through the circuit and the effect of high absolute humidity (AH) on the efficiency of the silicon detector (electronic supplementary material, table S1):

$$^{220}\text{Rn} = \frac{^{220}\text{Rn}_M}{D} \times \text{AH}, \tag{2.1}$$

where $D$ is equal to 0.1375, as a result of (i) the air flow rate through the circuit (0.8 l min$^{-1}$), (ii) the volume of the experimental circuit placed upstream of the Rn monitor (2.53 l), and (iii) the [220]Rn decay constant (0.756 min$^{-1}$). The mass water molecules hosted in the inner volume of RAD7 ($g$H$_2$O) may also neutralize part of the Rn daughters ([218]Po ions), lowering the efficiency of the silicon detector. The corrected value of AH (electronic supplementary material, table S1) is derived by the expression [48]:

$$\text{AH} = -m \times g\text{H}_2\text{O} + c, \tag{2.2}$$

where $m$ and $c$ are the slope and the intercept of linear regression fit of efficiency data plotted versus $g$H$_2$O, respectively [48].

## 2.4. Petrophysical analysis

Rock petrophysical properties and microstructural characterization were performed at the HP-HT Laboratory of the INGV. Ultrasonic $P$-wave ($V_p$) and $S$-wave ($V_s$) velocities and porosity were measured on both intact and thermally stressed rocks by cutting three cubes (3 cm/side) of material from the upper, central and lower portions of each cylindrical sample, in order to calculate a representative average value (electronic supplementary material, table S2). $V_p$ and $V_s$ were acquired using a high-voltage (1000 V) pulse generator and a Tektronix DPO4032 oscilloscope and two piezoelectric transducer crystals (100 kHz to 1 MHz frequency). The value of porosity ($\phi$) was determined using a He pycnometer (AccuPyc II 1340) with ±0.01% accuracy (electronic supplementary material, table S2).

The microstructural analysis was carried out by an optical petrographic microscope and a Jeol-JSM6500F field emission gun scanning electron microscope (FE-SEM) equipped with an energy-dispersive X-ray microanalysis system. High-magnification FE-SEM microphotographs were acquired in the back-scattered electron (BSE) imaging mode, whereas low-magnification optical photomicrographs were acquired under cross-polarized and parallel-polarized light.

X-ray powder diffraction (XRPD) patterns were collected with a Siemens D5005 diffractometer operating in the $\theta$–$2\theta$ vertical configuration, equipped with Ni-filtered CuK$\alpha$ radiation, installed at the Department of Ingegneria and Geologia of the University G. d'Annunzio. Each XRPD spectrum was recorded at $2\theta$ angles between $4°$ and $80°$, with a step scan of $0.02°$ and a counting time of 8 s (electronic supplementary material, table S3). XRPD patterns were first qualitatively evaluated to identify crystalline phases by a search-match comparison with the commercial inorganic crystal structure database. The crystalline phases that best match XRPD patterns, i.e. better reproduce positions and intensity of observed Bragg reflections, were then used as input crystallographic data.

# 3. Thermally induced physical and chemical changes

From a mineralogical point of view, the thermal treatment conducted from 100 to 900°C did not substantially change the phase assemblage of the rock either at the phenocryst or the matrix scale. The comparison between XRPD spectra obtained from the intact rock, EXP1 and EXP3 shows that the position and intensity of the mineral peaks are always equivalent (figure 1). Moreover, XRPD data do not reveal the appearance or disappearance of any crystalline phase during Rn thermal experiments (figure 1), in agreement with the high-thermal stability of the anhydrous igneous paragenesis [49,50]. Note that, at ambient pressure and under anhydrous conditions (such as those from this study), leucite, clinopyroxene and magnetite melt congruently at temperatures of 1391–1686°C; [51,52], much higher than those investigated in this study.

The intact rock sample is massive in hand specimen (figure 2a) and is composed of a dense polymineralic aggregate with no apparent discontinuities or fractures visible even under the microscope (figure 2b). The only exception is represented by several microcracks observable by BSE imaging at the micrometre to submicrometre scale (figure 2b). These small inter- and intra-crystal microcracks are typical of natural effusive products due to the effects of heat and gas loss during lava flow and emplacement. The value of $\phi$ (3.6%) is relatively low (electronic supplementary material, table S2), whereas $V_p$ (5.51 km s$^{-1}$) and $V_s$ (3.84 km s$^{-1}$) are in the range documented for rocks formed in massive lava flows [21,39]. On the other hand, EXP1 and EXP2 exhibit textural and petrophysical features different from those observed for the pristine sample. The applied heating and cooling rates (i.e. $\Delta T/\Delta t \gg 1°C/min$) are fast enough to induce significant thermal gradients through the rock sample [47,53]. This thermal stress leads to the opening of cracks accompanied by an increase in $\phi$ and a decrease in $V_p$ and $V_s$, as generally reported for massive igneous rocks [47,53–56]. Several fractures are visible by eye in hand specimen (figure 2c), as well as an almost continuous pattern of microcracks observable at the micrometre scale (figure 2d). Within the analytical uncertainty, the petrophysical properties measured for EXP1 and EXP2 are almost identical, corresponding to 5.2 and 5.4% $\phi$, 5.38 and 5.42 km s$^{-1}$ $V_p$ and 3.63 and 3.68 km s$^{-1}$ $V_s$, respectively (electronic supplementary material, table S2). The comparable textural and petrophysical features indicate that a similar level of microcrack damage was imparted to the rock sample during EXP1 and EXP2, either by using temperature steps (100°C $\Delta T$) up to 900°C or by applying a restricted number of heating/cooling cycles (300 and 600°C $\Delta T$). Leucite, clinopyroxene and magnetite show only limited thermal expansion at subvolcanic temperatures [57]. Moreover, a low number of heating/cooling cycles may not significantly enhance the natural crack patterns inherited from the time of eruption during the emplacement of the lava and its cooling at the surface [21,54].

By contrast, in EXP3 and EXP4, the rock sample was exposed to greater temperature changes attained by applying 20 heating/cooling cycles and a strong thermal gradient (800°C $\Delta T$). Millimetre-sized macroscopic faults progressively developed on the sample surface with increasing frequency of heating/cooling cycles (figure 3a). At the end of the thermal treatment, the phonolite was disaggregated into seven large rock fragments (figure 3b) and a pervasive network of a millimetre to micrometre cracks characterizes the rock sample (figure 3c). Owing to the thermal stress and the resulting microfracture network, the value of $\phi$ increases up to 8.2% (electronic supplementary material, table S2). Conversely, $V_p$ and $V_s$ decrease to 5.22 and 3.41 km s$^{-1}$, respectively, as a proxy for the enhanced thermal damage (electronic supplementary material, table S2). Therefore, both the high number of heating/cooling cycles and the strong thermal gradient caused rapid thermal microfracturing observable only at 800°C $\Delta T$. The intense rock failure is a consequence of the thermal stress imparted by expansion or contraction of the mineral grains up to levels corresponding to the tensile or shear strength of the phonolitic rock [55,58]. Although the igneous minerals in the phonolite

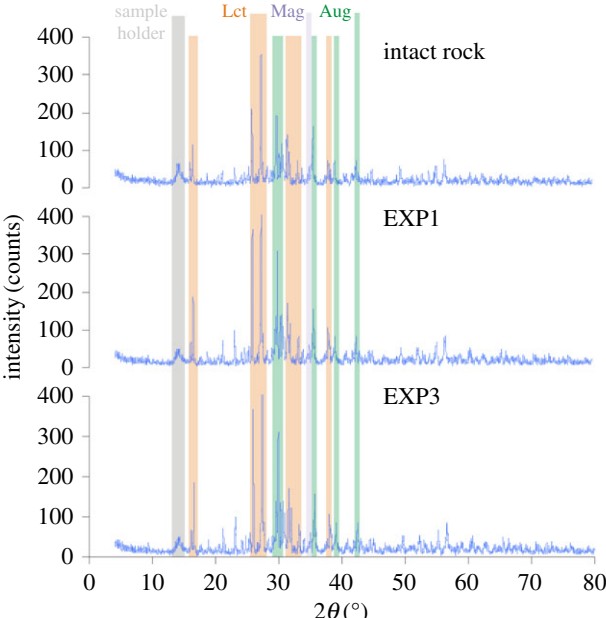

**Figure 1.** X-ray powder diffraction (XRPD) spectra tracking the stability of minerals in the intact phonolitic rock and during thermal experiments (EXP1 and EXP3). Aug, augite; Lct, leucite; Mag, magnetite.

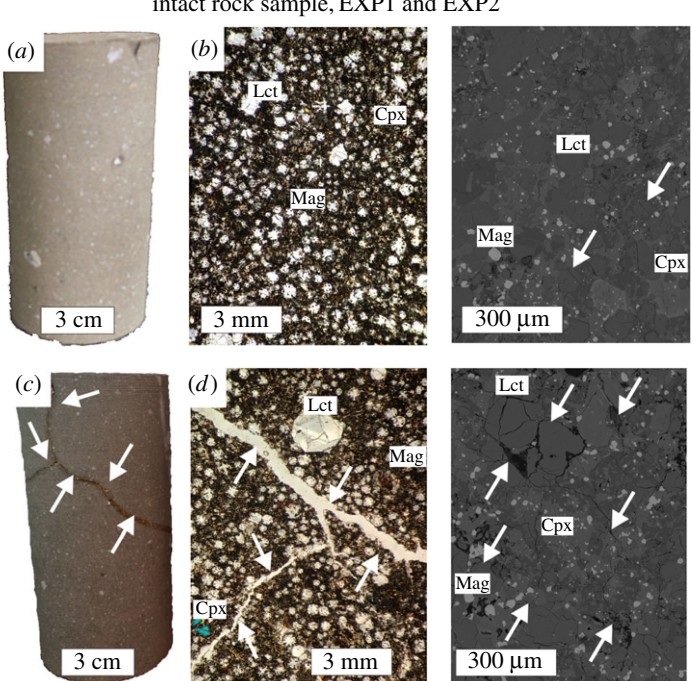

**Figure 2.** Representative structural and textural features of the intact rock sample (*a,b*) and experimental samples from EXP1 and EXP2 (*c,d*). The cylindrical rock sample is 50 mm in diameter and 110 mm in length. Microphotographs obtained by optical microscopy and FE-SEM are acquired at the millimetre and micrometre scale, respectively. The arrows indicate both microcracks and macroscopic fractures. Aug, augite; Lct, leucite; Mag, magnetite.

undergo only very limited structural change during heating, their isotropic thermal expansion differs as a function of the mineral species. According to previous authors [24,30,31,54], extremely fast heating/ cooling cycles (i.e. 5 min $\Delta t$ and 800°C $\Delta T$) lead to rapid thermal expansion mismatches of the different mineral constituents, thus instigating extensive microfracturing.

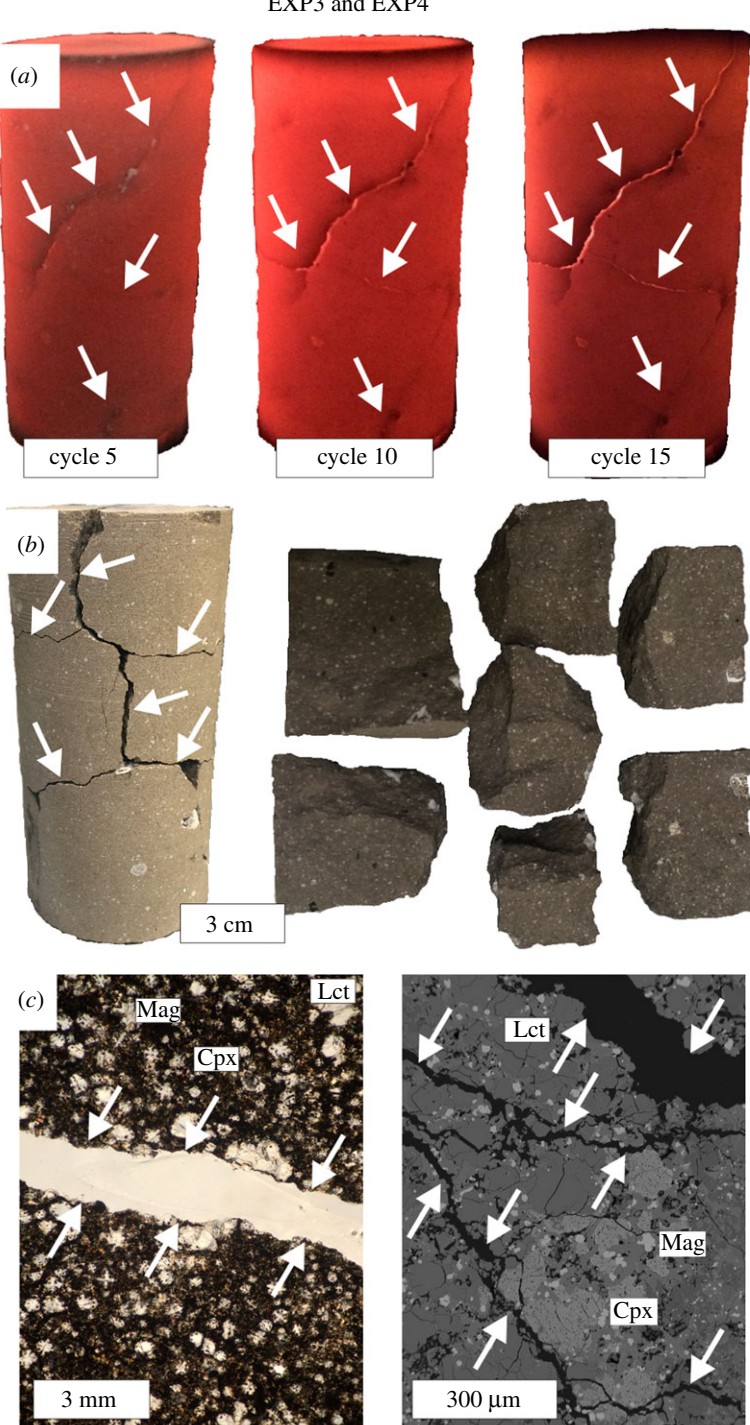

**Figure 3.** Representative structural and textural features of the rock sample from EXP3 and EXP4 obtained during intense heating/ cooling cycles (*a*), at the end of thermal stress experiment (*b*), and at the millimetre and micrometre scale by optical microscopy and FE-SEM, respectively (*c*). The arrows indicate both microcracks and macroscopic fractures. Cpx, clinopyroxene; Lct, leucite; Mag, magnetite.

## 4. Changes in the radon signal during thermal experiments

Owing to the stochastic process of radioactive decay, the $^{220}$Rn versus $t$ diagram (figure 4*a*) shows that, for each constant temperature, the Rn signal may deviate from the average value within a range of approximately 20–50%, in response to the statistical fluctuations of $^{220}$Rn measurements over the annealing time. Since the recoil effect is practically temperature independent [59], the role of

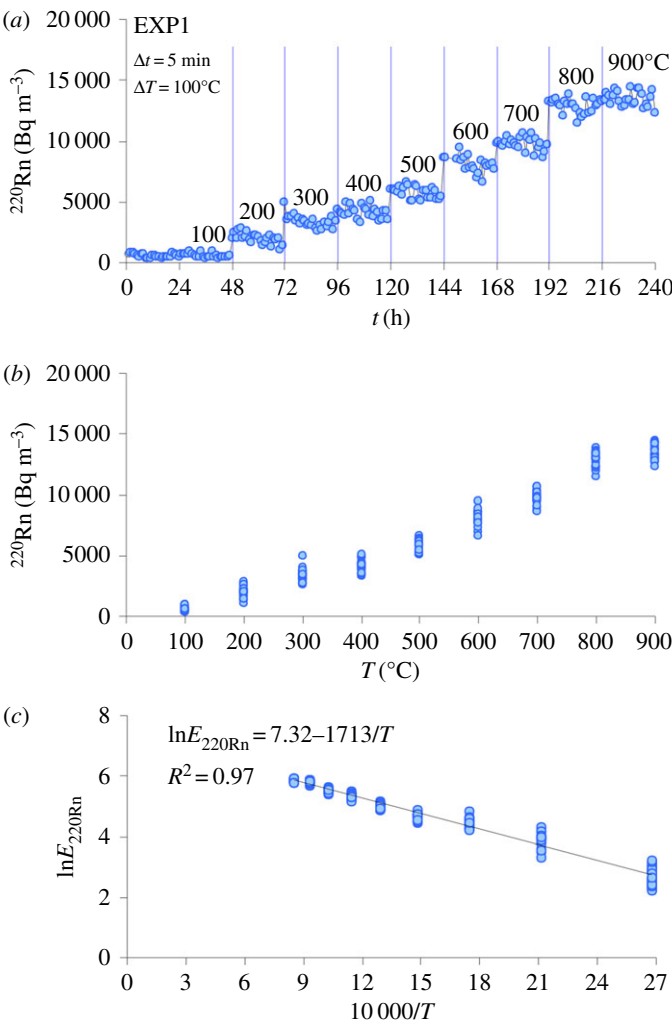

**Figure 4.** Results from EXP1 plotted in $^{220}$Rn versus $t$ (a), $^{220}$Rn versus $T$ (b) and $E_{220Rn}$ versus 10 000/$T$ (c) diagrams. The rock sample was first kept at 100°C for 48 h. Then, through a heating ramp ($\Delta t$) of 5 min, the temperature ($\Delta T$) was increased by 100°C and maintained constant for 22 h. This incremental step was replicated eight times (5 min $\Delta t$ and 100°C $\Delta T$) up to the maximum experimental temperature of 900°C. See Mollo *et al.* [29] for a detailed mathematical treatment of the nonlinear Arrhenius equation describing the dependence of Rn emission rate on temperature.

temperature is to increase the number of effective collisions of Rn atoms with other molecules and grain boundaries within the rock, thus favouring the migration of Rn gas through the microcracks of the phonolite [37,38]. Coherently, EXP1 shows that $^{220}$Rn monotonically increases by about two orders of magnitude as the temperature increases from 100 to 900°C (figure 4b). This behaviour corroborates the strong dependence of Rn diffusion on temperature, leading to transient Rn emissions due to a new value of equilibrium activity concentration [29,59,60]. The change in Rn emission rate ($E$ in arbitrary units) is positively correlated with the temperature-dependent diffusion coefficient of Rn by a nonlinear Arrhenius equation (where $T$ is expressed in Kelvin; [61–63]. From an empirical point of view, the plot ln$E$ versus $1/T$ (figure 4c) provides a quantitative description of the temperature-dependent migration of Rn atoms through the microcracks of the rock sample (see [29] for a detailed mathematical treatment). The regression fit of experimental data from this study (figure 4c) yields a high correlation coefficient ($R^2 = 0.97$), attesting to the close relationship between the enhancing effect of temperature and the escape of Rn atoms from the interconnected fractures of the rock sample.

Results from EXP2 confirm that $^{220}$Rn is significantly controlled by temperature changes, showing either increasing or decreasing values as a function of the heating/cooling cycles applied to the phonolite (figure 5a). The measurement of a transient Rn signal confirms the close relationship between temperature and the diffusion of gaseous radionuclides through the rock's structural irregularities which serve as Rn diffusion paths [59,62]. During these thermal cycles, the value of

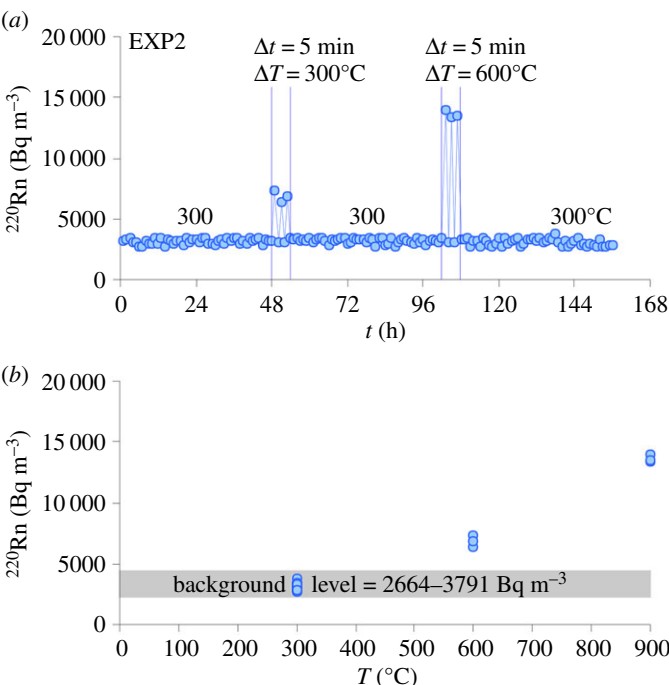

**Figure 5.** Results from EXP2 plotted in $^{220}$Rn versus $t$ (a) and $^{220}$Rn versus $T$ (b) diagrams. The rock sample was first kept at 300°C for 48 h. Then, the rock sample was subjected to low- and high-thermal stress cycles. For the low-temperature stress cycles (300°C $\Delta T$), the rock sample was heated from 300 to 600°C (5 min $\Delta t$) and, after 1 h, was cooled down to 300°C at the same rate. At the end of the third cycle, the temperature was maintained constant at 300°C for 48 h. The same strategy was applied for the high-temperature thermal stress cycles (600°C $\Delta T$) by increasing the temperature from 300 to 900°C.

$^{220}$Rn monitored for each single low-temperature (300°C $\Delta T$ and approx. 3200 Bq m$^{-3}$) and high-temperature (600°C $\Delta T$ and approx. 6800 Bq m$^{-3}$) cycle are almost comparable to that recorded during EXP1 (figure 4a), despite the different thermal treatments adopted. Moreover, the background level of Rn recorded at the end of the heating/cooling cycles of EXP2 falls into a restricted range (approx. 2600–3800 Bq m$^{-3}$, figure 5b), suggesting that the amount of microcrack damage imparted by the thermal gradients is not sufficient to cause detectable $^{220}$Rn modifications. According to Mollo *et al.* [39], no significant Rn changes are recorded in the laboratory during the deformation of low-porosity and high-strength phonolitic rocks. Most of the Rn atoms remain entrapped within the inter- and intra-granular microcracks, until the formation of macroscopic faults acting as new exhalation surfaces for the Rn gas [64]. The limited microcrack pattern developed in the thermally stressed phonolite (figure 2c,d) does not detectably change the number of diffusion paths for gas release. The development of microfracture networks in volcanic rocks depends strongly on the effect of rapid temperature changes in opening new degassing paths [47]. The comparison between figures 4b and 5b indicates that the Rn signal from EXP2 is primarily controlled by the temperature dependence of the Rn emission rate, thus resembling the values of $^{220}$Rn measured at 300, 600 and 900°C during EXP1.

In the EXP3, the thermal gradient condition caused by 20 heating/cooling cycles with 800°C $\Delta T$ is strong enough to substantially increase the number of diffusion paths and/or surface area, by developing a pervasive crack network (figure 3). The $^{220}$Rn versus $t$ diagram (figure 6a) shows that the Rn signal (approx. 6500 Bq m$^{-3}$) monitored for 120 h at 300°C is remarkably higher than for the equivalent temperature of 300°C during EXP1 (approx. 3500 Bq m$^{-3}$) and EXP2 (approx. 3400 Bq m$^{-3}$). The release of gas trapped in the microstructural fabric of the rock is constant over time due to the attainment of a steady-state condition (figure 6a) corresponding to a stationary Rn emission [38]. The background level of Rn (approx. 5900–7600 Bq m$^{-3}$, figure 6b) measured during EXP3 is increased by approximately 110% with respect to that monitored during EXP2 (figure 5b). The intense rock fracturing enhances the number of diffusion paths [43], attesting that the level of microcrack damage imparted on the phonolite is sufficient to drive changes in Rn release [65]. According to Sengupta *et al.* [66], the flux of $^{220}$Rn radionuclides from the fractured rock is proportional to its microstructural fabric where microfractures act as new exhaling surfaces and pathways for migration of Rn atoms. The fractures serve as channels to connect the isolated microcracks to the crystalline rock network. These migration channels develop in the rock structure and connect the isolated microcracks to the

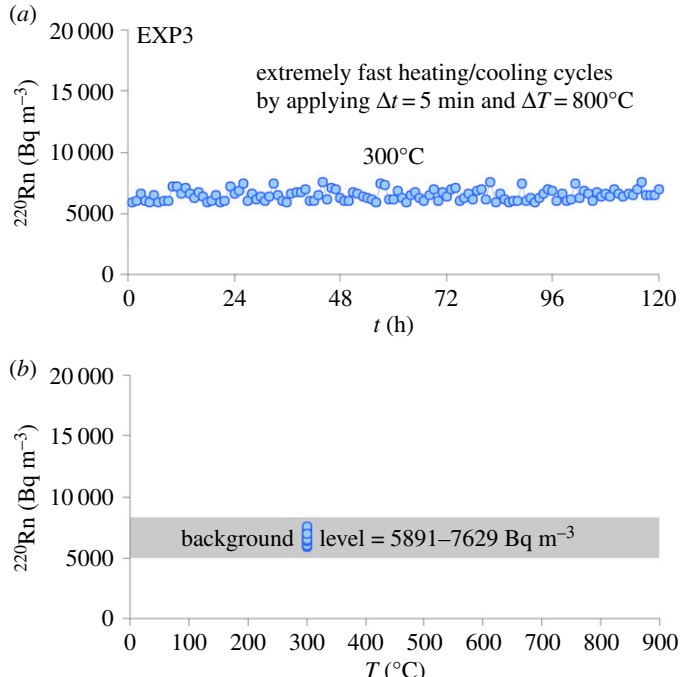

**Figure 6.** Results from EXP3 plotted in $^{220}$Rn versus $t$ (*a*) and $^{220}$Rn versus $T$ (*b*) diagrams. More intense heating/cooling cycles were performed with the aim to thermally stress the rock sample and to induce microfracturing. The rock sample was heated from 100 to 900°C (5 min $\Delta t$) and was then cooled from 900 to 100°C at the same rate. This heating/cooling cycle (5 min $\Delta t$ and 800°C $\Delta T$) was repeated 20 times. Owing to the short acquisition time, the Rn signal was not monitored during the fast heating/cooling steps. However, at the end of the twentieth cycle, the temperature of the furnace was decreased down to 300°C and maintained constant for 120 h, in order to measure the Rn change.

external rock surface, so that Rn liberated through interconnected fractures can easily migrate in the monitoring system [64].

The intense thermal microfracturing condition attained in EXP3 (i.e. rapid expansion/contraction of the rock grains) caused sample rupture by the formation of seven large rock fragments along the principal millimetre-sized macroscopic faults (figure 3*b*). The Rn signal emitted from these rock fragments is measured during EXP4 for 120 h at the constant temperature of 300°C (figure 7*a*). The presence of macroscopic rupture surfaces dramatically intensifies Rn emission, due to an enhanced migration of $^{220}$Rn atoms that escape more efficiently from the rock fragments [37,38]. The stochastic radioactive decay is accompanied by strong statistical fluctuations in $^{220}$Rn emissions, with deviations from the average value in the range of approximately 50–150%. Moreover, the background level of Rn (approx. 5900–19 900 Bq m$^{-3}$, figure 7*b*) increases up to approximately 420% with respect to that measured in EXP2 for the temperature dependence of the Rn emission rate. Therefore, it is concluded that the main effect of intense heating/cooling cycles on the phonolitic rock is that crack growth develops rapidly into macrofractures, thus forming macroscopic faults [55]. The enlarged surface area of the phonolite sample (figure 3*b*) results in a new value of equilibrium activity concentration of Rn and the increased fraction of accessible $^{220}$Rn atoms scales with the rock exhalation surface [39]. Since the Rn signal is proportional to the exhaling surface area, phonolites with relative small matrix minerals may also expose many grain boundary surfaces, thus significantly enhancing Rn emission rates [64].

# 5. Concluding remarks

We have investigated the effect of temperature on the Rn signal emitted from a phonolitic rock exposed to variable thermal conditions in the range of 100–900°C. Our experimental data indicate that a transient Rn signal is produced by the temperature-dependent diffusion of radionuclides through the rock's structural irregularities which serve as preferential gas pathways. However, this condition is attained only when thermal gradients do not induce substantial modifications in the rock structure and a relatively low

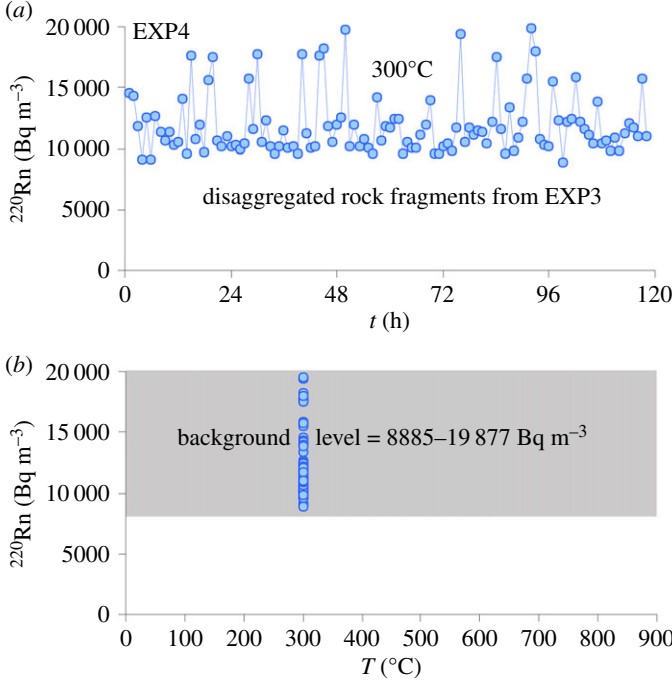

**Figure 7.** Results from EXP4 plotted in $^{220}$Rn versus $t$ ($a$) and $^{220}$Rn versus $T$ ($b$) diagrams. The thermally stressed rock sample obtained from EXP3 was disaggregated along the main macroscopic fault planes caused by the extremely fast heating/cooling cycles. Seven large rock fragments with centimetric dimensions were recovered. All the rock fragments were placed in the furnace preheated at 300°C and then the Rn signal was monitored for 120 h.

number of new cracks is introduced by the increasing temperature. In contrast, intense heating/cooling cycles can dramatically alter the petrophysical properties of the rock, by generating microfractures through thermal expansion and contraction of minerals. The diffusive microfracturing and the formation of macroscopic faults produce new exhalation surfaces that enhance the Rn signal up to a stationary level that is much higher than the temperature-dependent transient changes. Because of continuous $^{220}$Rn measurements in a closed-loop configuration, our experimental data are not directly comparable to natural $^{222}$Rn emissions at subvolcanic temperatures. However, the relative change in Rn signals observed in the laboratory can explain the origin of positive anomalies recorded on active volcanoes. Variations in Rn concentrations in volcanic areas account for an increased heat flow and release of volatiles (mainly $H_2O$ and $CO_2$) of magmatic origin, carrying Rn atoms from deep sources toward the surface [1–10]. Our results apply, however, to dyke injections generating thermal stress and deformation in the host rock. Expansion and contraction of the material adjacent to the dyke may cause aseismic deformation and microfracturing of subsurface rocks over long distances [11–13]. Positive Rn anomalies, such as those documented in this study, can be measured away from the location of the magmatic intrusion due to thermal stress propagation and opening of new pathways for the release of magmatic volatiles [11–13]. Future experimental studies should be designed to investigate and quantify the combined effects of heating/cooling cycles and variable carrier gas fluxes on Rn fluctuations. Results would be important to better interpret volcanic field monitoring data related to persistent injections of magma at very shallow depths and recording spatially heterogeneous Rn anomalies that are non-stationary in time.

Data accessibility. The experimental data of this article have been uploaded as electronic supplementary material.

Authors' contributions. S.M. and P.T. designed and performed the radon experiments. G.G. performed the experiments and applied correction to the radon signal. G.I. performed the mineralogical and petrographycal analyses. M.S. and P.S. contributed to writing the manuscript. All the authors give final approval for publication.

Competing interests. The authors declare no competing interests.

Funding. No research grant or funding agency supported this research.

Acknowledgements. Authors would like to thank Martina Mattina for her the help provided during laboratory activities. A. Schmitt (Associate Editor) and J. Blundy (Subject Editor) are acknowledged for their editorial work. An early version of this work benefited from the constructive and valuable comments of three anonymous reviewers. Thanks are also due to J. Caulfield for undertaking the correction of the English text.

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
