## [Reviewer comments · Royal Society Open Science]

Review History

RSOS-190782.R0 (Original submission)

Review form: Reviewer 1

Is the manuscript scientifically sound in its present form?

No

Are the interpretations and conclusions justified by the results?

Yes

Is the language acceptable?

Yes

Do you have any ethical concerns with this paper?

No

Have you any concerns about statistical analyses in this paper?

No

Recommendation?

Reject

Comments to the Author(s)

Please see the attached pdf file (Appendix A).

Review form: Reviewer 2 (Payton Gardner)

Is the manuscript scientifically sound in its present form?

Yes

Are the interpretations and conclusions justified by the results?

Yes

Is the language acceptable?

Yes

Do you have any ethical concerns with this paper?

No

Have you any concerns about statistical analyses in this paper?

No

Recommendation?

Accept with minor revision (please list in comments)

Comments to the Author(s)

Please see attached comments and annotated pdfs (Appendices B & C).

Review form: Reviewer 3

Is the manuscript scientifically sound in its present form?

Yes

Are the interpretations and conclusions justified by the results?

Yes

Is the language acceptable?

No

Do you have any ethical concerns with this paper?

No

Have you any concerns about statistical analyses in this paper?

No

Recommendation?

Major revision is needed (please make suggestions in comments)

Comments to the Author(s)

A very interesting paper:

a general wording comment: micrometric should be micrometer, etc throughout the paper. the paper should be rewritten with improved sentence structure.

the figure captions need to be rewritten, perhaps this will be done by the editors.

What do you anticipate the effect of pressure from depth to have on the processes observed here at ambient pressure?

Please comment on the fine grained nature of the rock studied, in view grain size studies of thermal cracking; i.e. from surface area considerations one would not expect a great deal of thermal cracks from differential thermal expansion of mineral grains: how does this help or hinder the processes you are studying?

line:

60-reword

62: redundant

77: what does highly crystalline rock mean?

92: what does characteristically brittle mean, please reference

102: i could not access reference [28] to read about the experimental setup, and we have had experience with many rock heating studies: a question arises as to how one can heat a sample of this size at the rates stated, with essentially no over run of temperature? please give enough detail in the paper to explain this--it ultimately may not matter given the results, but this reviewer would like to see this detail.

201: micrometer

221- please explain "more efficient thermal stress"

230- some believe that thermal shock is faster than the heating/cooling in these experiments

242- delete "of"

270/1- reword

290- thermal shock condition- suggest rethink using this terminology

301- refs 66,67 refer to mechanically stress induced fractures and faulting, and not thermally induced fracture, so these references should be deleted.

308--reword to "concluding"

309/310- reword from "highly crystalline"

Was any quantitative work done with the observational studies beyond that presented?

The work is predicated on 4 tests, while this represents a lot of information on these tests, they are 4 different tests. how are generalities derived?

There should be more to the "conclusions" section, please expound a bit.

Decision letter (RSOS-190782.R0)

26-Jul-2019

Dear Silvio,

The editors assigned to your paper ("Transient to stationary radon emissions from highly

crystalline rocks exposed to subvolcanic temperatures in laboratory") have now received comments from reviewers. We would like you to revise your paper in accordance with the referee and Associate Editor suggestions which can be found below (not including confidential reports to the Editor). Please note this decision does not guarantee eventual acceptance. The referee comments and decision of the Associate Editor are quite substantial, but also constructive. I am confident that they will help you to produce a better, clearer and more impactful paper, if you take into account these comments and address them fully in a revised version.

Please submit a copy of your revised paper before 18-Aug-2019. Please note that the revision deadline will expire at 00.00am on this date. If we do not hear from you within this time then it will be assumed that the paper has been withdrawn. In exceptional circumstances, extensions may be possible if agreed with the Editorial Office in advance. We do not allow multiple rounds of revision so we urge you to make every effort to fully address all of the comments at this stage. If deemed necessary by the Editors, your manuscript will be sent back to one or more of the original reviewers for assessment. If the original reviewers are not available, we may invite new reviewers.

- Data accessibility

If you wish to submit your supporting data or code to Dryad (<http://datadryad.org/>), or modify your current submission to dryad, please use the following link:
<http://datadryad.org/submit?journalID=RSOS&manu=RSOS-190782>

- Competing interests

- Authors' contributions

- Acknowledgements

- Funding statement

on behalf of Professor Axel Schmitt (Associate Editor) and Jon Blundy (Subject Editor)
openscience@royalsociety.org

Associate Editor's comments (Professor Axel Schmitt):

Dear Dr. Mollo,

Your manuscript has now been reviewed by three experts in the field. As you can see, a good measure of criticism has been voiced, and although one of the reviewers recommends rejection, I feel that even this most critical review recognizes the potential of your research. However, there is clearly a need for substantial changes before acceptance can be considered.

For making these changes, you will find many helpful comments in the reviews, and I concur with most of them. I would like to reinforce that the current title is misleading: most petrologists will not associate the rock that you used in your experiments with the qualifier "highly crystalline" as the title suggests. A more specific and adequate title is therefore required.

The manuscript has many deficiencies in the use of standard English. In addition to grammatical and semantic errors pointed out by the reviewers, there are often awkward and unnecessary complicated phrases. For example in the abstract, please consider potential corrections for:

“thin-to-thick” (of variable thickness)

“lithological formations” (rocks)

“physicochemical features” (physical and chemical properties)

“In this respect” (To assess radon mobilization in subvolcanic thermal regimes, we have performed ...)

“However, it has been also observed that” (not needed)

“produces” (produced)

“intensifies dramatically” (reverse, as suggested by reviewer, and state “intensified”, as you are still describing your experiments)

“giving reason for” (which explains that ...)

There should be a much clearer concluding sentence at the end of the abstract what the implications of your research are, especially addressing a broader audience.

Regards,

Axel Schmitt (Heidelberg University)

Comments to Author:

Reviewers' Comments to Author:

Reviewer: 1

Comments to the Author(s)

Please see the attached pdf file.

Reviewer: 2

Comments to the Author(s)

Please see attached comments and annotated pdf.

Reviewer: 3

Comments to the Author(s)

A very interesting paper:

a general wording comment: micrometric should be micrometer, etc throughout the paper.

the paper should be rewritten with improved sentence structure.

the figure captions need to be rewritten, perhaps this will be done by the editors.

What do you anticipate the effect of pressure from depth to have on the processes observed here at ambient pressure?

Please comment on the fine grained nature of the rock studied, in view grain size studies of thermal cracking; i.e. from surface area considerations one would not expect a great deal of thermal cracks from differential thermal expansion of mineral grains: how does this help or hinder the processes you are studying?

line:

60-reword

62: redundant

77: what does highly crystalline rock mean?

92: what does characteristically brittle mean, please reference

102:i could not access reference [28] to read about the experimental setup, and we have had experience with many rock heating studies: a question arises as to how one can heat a sample of this size at the rates stated, with essentially no over run of temperature? please give enough detail in the paper to explain this--it ultimately may not matter given the results, but this reviewer would like to see this detail.

201:micrometer

221- please explain "more efficient thermal stress"

230-some believe that thermal shock is faster than the heating/cooling in these experiments

242- delete "of"

270/1- reword

290-thermal shock condition-suggest rethink using this terminology

301-refs 66,67 refer to mechanically stress induced fractures and faulting, and not thermally induced fracture, so these references should be deleted.

308--reword to "concluding"

309/310- reword from "highly crystalline"

Was any quantitative work done with the observational studies beyond that presented?

The work is predicated on 4 tests, while this represents a lot of information on these tests, they are 4 different tests. how are generalities derived?

There should be more to the "conclusions" section, please expound a bit.

Author's Response to Decision Letter for (RSOS-190782.R0)

See Appendix D.

Decision letter (RSOS-190782.R1)

27-Aug-2019

Dear Dr Mollo:

Manuscript ID RSOS-190782.R1 entitled "Transient to stationary radon (^{220}Rn) emissions from a phonolitic lava exposed to subvolcanic temperatures" which you submitted to Royal Society Open Science, has been reviewed. The comments of the Editor are included at the bottom of this letter.

Please submit a copy of your revised paper before 19-Sep-2019. Please note that the revision deadline will expire at 00.00am on this date. If we do not hear from you within this time then it will be assumed that the paper has been withdrawn. In exceptional circumstances, extensions may be possible if agreed with the Editorial Office in advance. We do not allow multiple rounds of revision so we urge you to make every effort to fully address all of the comments at this stage. If deemed necessary by the Editors, your manuscript will be sent back to one or more of the original reviewers for assessment. If the original reviewers are not available we may invite new reviewers.

To revise your manuscript, log into <http://mc.manuscriptcentral.com/rsos> and enter your

Author Centre, where you will find your manuscript title listed under "Manuscripts with Decisions." Under "Actions," click on "Create a Revision." Your manuscript number has been appended to denote a revision. Revise your manuscript and upload a new version through your Author Centre.

- Ethics statement

- Data accessibility

- Competing interests

- Authors' contributions

- Acknowledgements

- Funding statement

Kind regards,

Alice Power

Editorial Coordinator

on behalf of Professor Axel Schmitt (Associate Editor) and Jon Blundy (Subject Editor)

Associate Editor Comments to Author (Professor Axel Schmitt):

The revised version has now accommodated many of the reviewer's suggestions, and in some instances where the authors did not follow the reviewer's criticisms, this was justified for the sake of brevity and focus. Thus overall the manuscript now succeeds in adequately summarizing and discussing a set of interesting experiments and their potential implications for Rn monitoring. However, there are still many shortcomings in the presentation that either are due to deficiencies in the English, or the frequent use of jargon, often obscuring the meaning of the sentences. I have made significant edits in the attached document to improve the manuscript, but there is clearly a need for another careful iteration by the authors, preferably also involving proof-reading by a native speaker, to correct the English and bring the manuscript to an acceptable standard.

Editorial Office Comments to Authors:

For information about language editing services endorsed by the Royal Society, please follow the link below:

<https://royalsociety.org/journals/authors/language-polishing/>

Author's Response to Decision Letter for (RSOS-190782.R1)

See Appendix E.

Decision letter (RSOS-190782.R2)

03-Oct-2019

Dear Dr Mollo,

I am pleased to inform you that your manuscript entitled "Transient to stationary radon (^{220}Rn) emissions from a phonolitic rock exposed to subvolcanic temperatures" is now accepted for publication in Royal Society Open Science.

The editors consider the changes you have made to be satisfactory and have now recommended acceptance.

on behalf of Professor Axel Schmitt (Associate Editor) and Jon Blundy (Subject Editor)
openscience@royalsociety.org

Manuscript number: RSOS-190782

Title: Transient to stationary radon emissions from highly crystalline rocks exposed to subvolcanic temperatures in laboratory

Authors: S. Mollo, P. Tuccimei, M. Soligo, G. Galli, G. Iezzi, P. Scarlato

Review report

This article reports the gaseous measurement of the activity concentration of thoron (radon-220) released from a phonolitic rock subjected to heating events and heating/cooling cycles from 100°C to 900°C, characteristic temperatures that may result from magmatic intrusion underneath a volcanic edifice. A total of four experiments performed show that transient increases in thoron release occur during heating. The authors argue that temperature-dependent diffusion controls the release of thoron and that intense heating/cooling cycles form micro-fractures which interconnect, hence opening new pathways for thoron release.

The paper is generally easy to read and to follow. The figures also give a clear message to the reader. The general organization of the paper is acceptable. However, the points detailed below (major comments) deserve particular attention:

(1) The title and the introduction section are misleading as they only present radon, which generally corresponds to radon-222 isotope, and not to thoron (radon-220 isotope) which is measured in this study. I would suggest correcting this throughout the manuscript and in the title. In addition, one volcanic rock type is studied only, not several “highly crystalline rocks”. Finally, about 20% of cited articles in the reference list are self-citations which may appear a little too much.

(2) The introduction section is rather short and not quantitative enough. Several works have studied the effects of temperature and pressure on radon (less on thoron) on various lithologies, but a little only is recalled in the introduction section, although almost all these works are cited. I would suggest adding a paragraph relying on the known/identified/suspected effects of temperature on the release of radon and thoron from porous materials such as rocks in the laboratory, giving quantitative numbers. Below, I inserted some references that could be used in this new paragraph:

Garver, E., Baskaran, M., 2004. Effects of heating on the emanation rates of radon-222 from a suite of natural minerals. *Appl. Radiat. Isot.* 61, 1477-1485.

Girault, F., Perrier, F., 2011. Heterogeneous temperature sensitivity of effective radium concentration from various rock and soil samples. *Nat. Hazards Earth Syst. Sci.* 11, 1619-1626.

Iskandar, D., Yamazawa, H., Iida, T., 2004. Quantification of the dependency of radon emanation power on soil temperature. *Appl. Radiat. Isotopes* 60, 971-973.

Jobbágy, V., Somlai, J., Kovács, J., Szeiler, G., Kovács, T., 2009. Dependence of radon emanation of red mud bauxite processing wastes on heat treatment. *J. Hazard. Mater.* 172, 1258-1263.

Sakoda, A., Ishimori, Y., 2014. Calculation of temperature dependence of radon emanation due to alpha recoil. *J. Radioanal. Nucl. Ch.* 299, 2013-2017.

Sas, Z., Somlai, J., Szeiler, G., Kovács, T., 2012. Radon emanation and exhalation characteristic of heat-treated clay samples. *Radiat. Protect. Dosim.* 152, 51-54.

(3) The authors should differentiate transient release from steady-state release of thoron. In order to do that on a clearer manner, I would suggest to measure the thoron release from the samples before any heating experiment is performed and then to compare it with post-heating data. Not merely comparing equilibrium activity concentrations of thoron in the closed system, but instead to do that quantitatively, I would suggest calculating an effective emanation of thoron (or an effective radium-224 concentration), as it has been done routinely for radon (effective radium-226 concentration). This will give a physical value for the thoron emanating power of the studied samples. For example, the data plotted in Fig. 7, which gives thoron activity concentration at an

infinite time compared with thoron half-life, should be used to obtain a thoron emanating power of the sample. This more quantitative approach will also give information on the reversibility or irreversibility of the (micro-) fracturing induced by the heating events and heating/cooling cycles.

(4) The experiments carried out on the samples are worth of investigation. However, I would suggest associating every performed experiment to a natural phenomenon that may happen underneath a volcano, or elsewhere. I suspect that this point would necessitate new experiments, but this would deeply strengthen the scope of the paper.

(5) I am wondering how an apparatus dedicated to measure thoron activity concentration in the air under normal conditions is also able to measure it under high temperature (i.e. above normal operating temperature, which is generally below 45°C). At minimum, this will increase the noise level and probably the statistical dispersion of counts. Did you use any cooling procedure before the measurement in the apparatus? In addition, the sampling interval of one hour appears quite large compared with the thoron half-life of less than 1 min. For radon-222, a sampling interval of 10 min to 1 h is generally necessary to detect fast transient changes, as it has been done for example in this recent study:

Girault, F., Schubnel, A., Pili, É., 2017. Transient radon signals driven by fluid pressure pulse, micro-crack closure, and failure during granite deformation experiments. *Earth Planet. Sci. Lett.* 474, 409-418.

(6) The methodology does not include any uncertainty description. What are the sources of uncertainty and what is the experimental uncertainty for a given thoron activity concentration measurement?

All these major points imply that, according to me, your paper cannot be accepted at this stage. I encourage you to carry out new experiments, making sure all the points mentioned above are checked, and to resubmit a stronger manuscript to *Royal Society Open Science*, or elsewhere. I hope that the comments above and below will help you in this task.

Other comments:

- 1) Line 49: Probably geochemical anomalies in active volcanic settings have not been observed in Italy only.
- 2) Line 64: Please detail more what is the thermal weakening process and why the study of thoron should help.
- 3) Line 72: There are many other papers describing this effect, including increases of porosity and permeability.
- 4) Lines 74-75: The effect on this “emanating power” should be explained in a dedicated paragraph. What are the general effects? Please try to be more quantitative.
- 5) Line 76: If we take time to look at what has been done, generally we have data, but the understanding of the processes involved is poorly known. Building a model reproducing data is more difficult. Proposing a simple model together with your data-set to explain it would be extremely rewarding.
- 6) Line 92: “behaviour”.
- 7) Lines 103-105: This sentence should be moved to the introduction section.
- 8) Line 106: “under the following conditions”.
- 9) Lines 144-115: One hour may appear too much to measure thoron knowing its half-life of less than a minute.
- 10) Lines 123-124: I do not understand why it was not possible to measure thoron activity concentration continuously, while it has been done in the other experiments.
- 11) Line 130: This stationary effect you are analyzing here should be studied using a steady-state physical quantity, such as the effective radium-224 concentration.

- 12) Line 134: Please add the manufacturer's country.
- 13) Lines 137-139: One-hour sampling interval is also enough to measure radon-222 activity concentration.
- 14) Line 142: Please insert a reference to substantiate this assertion.
- 15) Line 150: Is there any leakage of the system? Did you measure it using dedicated experiments?
- 16) Line 165: Please remove the extra words "from the".
- 17) Line 178: "installed at the".
- 18) Lines 184-185: This sentence is redundant with the sentences written just above.
- 19) Lines 188-190: This is counter-intuitive and does not match former studies showing mineral transformation(s) during heating experiments. Actually, we do see a difference in Fig. 1 when looking at the two largest peaks of the leucite between 25° and 30° (2θ), with a significant increase of the first peak after heating to 900°C. Please check.
- 20) Line 196: I agree with that, but partial melting can nevertheless appear, especially at 900°C.
- 21) Lines 203 and 212: Please give uncertainties to appreciate the differences or similarities.
- 22) Lines 245-246: How is your quantity E calculated? Generally E is used as the emanation factor for radon and thoron. You may use ΔE as the emanation difference.
- 23) Line 246: Please do not use this adverb "sympathetically" here.
- 24) Lines 254-255: I agree with that point, but with the sampling time interval of 1 h, you cannot claim that the release of ^{220}Rn is instantaneous after heating because you cannot measure it experimentally.
- 25) Lines 258-259: It appears here irreversible indeed. Please check whether all the heating procedures you used imply irreversible changes in your samples.
- 26) Lines 269-270: Please rewrite this sentence.
- 27) Lines 271-273: Please try to be more quantitative.
- 28) Line 277: What was the background value before the heating experiment? A given block can also show some significant heterogeneity between its core samples.
- 29) Lines 278-280: This last experiment is relatively obvious and should not be discussed this way. Please see the main comment above.
- 30) Lines 287-289: This sentence is interesting, but does not give any new message to the reader as this has already been observed.
- 31) Lines 295-296: Why the statistical counting fluctuations are larger than in other experiments? Could it be the temperature of the gas entering the apparatus?
- 32) Line 302: Equilibrium activity concentration can be used to calculate a physical quantity.
- 33) Line 304: You did not measure the exhaling surface area. What is the diffusion length of thoron in a porous media? What was the size of the seven samples compared with the size of a core?
- 34) Lines 309-319: Generally, the conclusion section is different from the abstract. Please recall the main aspects of the paper and open some perspectives.
- 35) Line 357: "Heiligmann". Line 486: Please correct the alignment. Please check the reference list.
- 36) Fig. 1: Raw data are not so useful. Please quantify the changes.
- 37) Figs. 2 and 3: Please rewrite the caption describing the subfigures a, b, c, and d separately.
- 38) Fig. 4: The two steps of thoron concentration increase from 500 to 600°C and from 700 to 800°C are relatively larger than the others. Could you explain this observation? Is there any mineralogical transformation at these two temperatures?
- 39) Line 547: "At the end of the third cycle".
- 40) Figs. 5, 6 and 7: The subplot b does not appear essential.
- 41) Figs. 6 and 7: This figure should be removed after the calculation of a physical quantity describing the thoron emanating power of the samples is done.
- 42) Fig. 7: Were the samples entirely cooled during the measurement of thoron concentration?

Appendix B

Dear Editor,

I have read and reviewed “Transient to stationary radon emissions from highly crystalline rocks exposed to subvolcanic temperatures in laboratory” by Mollo et al. The paper described a set of experiments looking at the effect of temperature on rock radon exhalation. The results clearly show that temperature strongly controlled the rate of Rn production from whole rock cores, and that strong thermal cycling can cause enough thermal stress induced damage to increase the exhalation rate. I think this is an important result and worth of publication. The results appear to be real and the interpretation is scientifically sound. I have some minor editorial commentary designed to help the paper be clearer and more understandable, but recommend publication after these minor changes are made.

Concerns:

There are two basic steps of Rn release from the rock core.

- 1) In step one the Rn must be released from the crystal lattice into the adjacent pore space and fracture network. This release can happen via alpha recoil during Rn production, mechanical fracture of mineral grains and subsequent release, and to a lesser extent diffusion through the mineral grains. This step is likely dominated by alpha recoil, which is not temperature dependent.
- 2) In the second step, the Rn moves through the pore and fracture network to the outside of the core. If there is no pressure gradient, this process is dominated by diffusion which is highly temperature dependent.

I am fairly certain the authors' are mostly referring to process 2) when they discuss release results, but I think they should add even a few sentences to clarify the release process and which process they think is dominant.

I really think the authors should give at least a basic description of the analytical system rather than simply refer readers to another journal article. The article should stand by itself, and it is very hard to interpret laboratory results, without a basic understanding of the measurement apparatus.

I think the conclusions could use some general advice for future researcher concerning when and where thermal cycling driven radon release will be an important process to concern. These should be order of magnitude quantitative, but give readers some quantitative advice on when this process should be considered important.

The English is decent, but could definitely use some tightening up. I have attached an annotated pdf with some editorial suggestions, but it is by no means exhaustive.

I have attached an annotated pdf with 27 additional comments and editorial suggestions.

Appendix C**ROYAL SOCIETY
OPEN SCIENCE****Transient to stationary radon emissions from highly
crystalline rocks exposed to subvolcanic temperatures in
laboratory**

Journal:	Royal Society Open Science
Manuscript ID	RSOS-190782
Article Type:	Research
Date Submitted by the Author:	08-May-2019
Complete List of Authors:	Mollo, Silvio; Sapienza University of Rome Tuccimei, Paola; Roma Tre University Faculty of Mathematics Physics and Natural Science Soligo, Michele; Roma Tre University Faculty of Mathematics Physics and Natural Science Galli, Gianfranco; INGV Iezzi, Gianluca; Gabriele d'Annunzio University of Chieti and Pescara Scarlato, Piergiorgio; INGV
Subject:	Nuclear chemistry < CHEMISTRY, Volcanology < EARTH SCIENCES, Geochemistry < EARTH SCIENCES
Keywords:	radon emission, subvolcanic thermal gradients, temperature-dependent radon diffusion
Subject Category:	Earth science

**Author-supplied statements**

Relevant information will appear here if provided.

***Ethics***

*Does your article include research that required ethical approval or permits?:*

This article does not present research with ethical considerations

*Statement (if applicable):*

CUST_IF_YES_ETHICS :No data available.

***Data***

*It is a condition of publication that data, code and materials supporting your paper are made publicly*
*available. Does your paper present new data?:*

Yes

*Statement (if applicable):*

New data are provided as supplementary material

***Conflict of interest***

I/We declare we have no competing interests

*Statement (if applicable):*

CUST_STATE_CONFLICT :No data available.

***Authors' contributions***

This paper has multiple authors and our individual contributions were as below

*Statement (if applicable):*

S.M. and P.T. designed and performed the radon experiments. G.G. performed the experiments and
applied correction to the radon signal. G.I. performed the mineralogical and petrographical
analyses. M.S. and P.S. contributed to writing the manuscript. All the authors give final approval for
publication.

**Transient to stationary radon emissions from highly crystalline rocks exposed to**
**subvolcanic temperatures ~~in laboratory~~**

¹Silvio Mollo, ²Paola Tuccimei, ²Michele Soligo, ³Gianfranco Galli,

⁴Gianluca Iezzi, ⁵Piergiorgio Scarlato

¹ Dipartimento di Scienze della Terra, Sapienza - Università di Roma, P.le Aldo Moro 5, 00185
Roma, Italy

² Università “Roma Tre”, Dipartimento di Scienze, Largo S. L. Murialdo 1, 00146 Roma, Italy

³ Istituto Nazionale di Geofisica e Vulcanologia, Via di Vigna Murata 60, 00143 Roma, Italy

Gianfranco Galli, Piergiorgio Scarlato

⁴ Dipartimento di Ingegneria & Geologia, Università G. d'Annunzio, Via dei Vestini 30, 66013
Chieti, Italy

Corresponding author:

Silvio Mollo

Sapienza-Università di Roma

Dipartimento di Scienze della Terra

P.le Aldo Moro 5

00185 Roma, Italy

e-mail: silvio.mollo@uniroma1.it

Abstract

Rock substrates beneath active volcanoes are frequently subjected to temperature changes caused by the input of new magma from depth and/or the intrusion of thin-to-thick magma bodies within the subvolcanic lithological formations. The primary effect of the influx of hot magma is the heating of the surrounding host rocks with the consequent modification of their physicochemical features. In this respect, we have performed radon thermal experiments on a phonolitic rock exposed to temperatures in the range of 100-900 °C, with the aim to reproduce the most common subvolcanic thermal regimes. Results from these experiments indicate that transient radon signals are not unequivocally related to substrate deformation caused by tectonic stresses, but rather to a temperature-dependent diffusion of radionuclides through the structural discontinuities of the rocks which serve as preferential pathways for gas release. However, it has been also observed that intense heating/cooling cycles are accompanied by rapid expansion and contraction of minerals. This strong thermal shock condition produces both inter- and intra-crystal microfracturing, as well as the formation of macroscopic faults. The increased number of diffusion paths ~~intensifies~~ ~~dramatically~~ the radon atom migration, giving reason for a stationary background level that is invariably much higher than the temperature-dependent transient changes. 
Keywords: radon emission; subvolcanic thermal gradients; temperature-dependent radon diffusion

1. Introduction

There is growing recognition that radon monitoring represents an important study area for the investigation of premonitory signals in active volcanic settings and for interpreting geochemical anomalies before volcanic eruptions (e.g., [1-8]). The volcanic activity is commonly preceded from days to months by an increasing volcanic tremor and marked radon changes due to magmatic intrusions and deformations within the flanks of the volcanic edifice [9,10]. Small dyke injections may generate localised stress conditions along faults and fissures in the volcanic pile, causing marked radon anomalies [11]. Conversely, large volumes of intrusive magma cause substantial changes in the stress field within the subvolcanic rock substrate, so that radon anomalies can be measured over distances of several tens to hundreds of kilometres [12-14]. Thermal effects due to heat flow and increasing volcanic temperatures for months or even years are frequently accompanied by high radon emissions [15,16]. An intense thermal regime facilitates the extraction of radon from subvolcanic rocks and, sometimes, the increase of radon signal scales with the rate of magma uprising and gas release through the fractured rocks [17]. A transient state condition can also result from temperature gradients and carrier gases, causing ~~that~~ radon emissions are frequently spatially heterogeneous and non-stationary in time [11,18,19].

~~In this framework, subvolcanic rock substrates beneath active volcanoes around the world consist of~~ highly dynamic environments in which the geological materials are subjected to substantial physicochemical changes caused by injection of magmas from depth [20-24]. The ascending magma batches may stall at very shallow levels, even within the lava pile of the volcanic edifice, or may feed complex dike networks intruding at a few meters from the ground surface. An intense heat flow is released by the hotter magma bodies into the colder host rocks of the subvolcanic substrates [25] that, in turn, may undergo cyclic heating and cooling stages [26]. Heat conduction produces thermal gradients on the order of ~100-1000 °C that propagate along distances of thousands of meters, incorporating several cubic kilometers of subvolcanic rocks [27,28]. Thermal stresses within the crystalline lithologies produce expansion and contraction of the

72 constituent minerals, frequently forming a network of pervasive microfractures [29,30]. Despite a
73 gamut of experimental studies ~~have extensively investigated~~ the effect of rock deformation on radon
emission [31-41], the role played by subvolcanic thermal regimes on the emanating power of
volcanic piles and the background level of radon signal has received much less attention [28,42]. In
order to address this paucity of information, we have conducted radon thermal experiments on a
highly crystalline rock exposed to increasing temperatures and variable heating/cooling cycles.
Results from real-time radon monitoring in a thermal range of 100-900 °C allow ~~to~~ better
understand ~~and quantify~~ the relationship between radon signal and magmatic activity during
thermally-induced physicochemical changes of the subvolcanic rocks.

**2. Methods**

*2.1. Starting material*

A high strength crystalline lava from the Colli Albani volcanic district (Latium, Italy) ~~were~~
used for the radon thermal experiments (cf. [38]). The Colli Albani volcanic district is ~~nearby~~ the
city of Rome and its period of activity, from 608 to 36 ka, was characterised by effusive and
explosive eruptions fed by silica-undersaturated ultrapotassic magmas [43-45]. The rock selected
for the radon thermal experiments is a phonolite composed of millimetre- to submillimeter-sized
phenocrysts of leucite, clinopyroxene (augite) and magnetite in order of abundance, and
submillimeter-sized matrix minerals that are identical to the phenocryst assemblage. The
petrophysical properties of the phonolite have been already investigated by Mollo *et al.* [38]: the
deformation behavior is characteristically brittle, the total rock porosity (ϕ) is 3.6 %, and P-wave
(V_p) and S-wave (V_s) velocities are 5.51 and 3.84 km s⁻¹, respectively. For the purpose of this study,
the same block of phonolite was cored to obtain cylindrical rock samples 50 mm in diameter and
110 mm in length.
*2.2. Radon thermal experiments*

The experimental setup used to perform radon thermal experiments consists of a furnace
equipped with a radon monitoring system specifically designed and developed at the HP-HT
Laboratory of Experimental Volcanology and Geophysics of the Istituto Nazionale di Geofisica e
Vulcanologia (INGV) in Rome (Italy), in order to analyse the radon signal emitted from rocks
exposed to subvolcanic temperatures (see [28] for a detailed description of the experimental setup).
This work aims to better delineate the effects of background thermal gradients typical of active
volcanic areas on the radon emissions from crystalline rocks that compose the subvolcanic
basements. Time- and temperature-dependent variations of the radon signal have been investigated
through four different thermal experiments carried out at the following conditions:

- EXP1. The rock sample was first kept at 100 °C for 48 h. Then, through a heating ramp (Δt)
of 5 min, the temperature was increased (ΔT) by 100 °C and maintained constant for 22 h.

This incremental step was replicated eight times (5 min Δt and 100 °C ΔT) up to the
maximum experimental temperature of 900 °C;

- EXP2. The rock sample was first kept at 300 °C for 48 h and, then, was subjected to low-
and high-thermal stress cycles. For the low-temperature stress cycles (300 °C ΔT), the rock
sample was heated from 300 °C to 600 °C (5 min Δt) and, after 1 h, was cooled down to
300 °C at the same rate. Note that the dwell time of 1 h was sufficient to analyse the radon
signal [37] and to ensure thermal homogenisation through the rock sample [46]. At the end
of third cycle, the temperature was maintained constant at 300 °C for 48 h. The same
strategy was applied for the high-temperature thermal stress cycles (600 °C ΔT) by
increasing the temperature from 300 to 900 °C;

- EXP3. More intense heating/cooling cycles were performed with the aim to thermally stress
the rock sample and to induce microfracturing. The rock sample was heated from 100 to
900 °C (5 min Δt) and then was cooled from 900 to 100 °C at the same rate. This
heating/cooling cycle (5 min Δt and 800 °C ΔT) was repeated twenty times. Due to the short

acquisition time, the radon signal was not monitored during the very fast heating/cooling
steps. However, at the end of the twentieth cycle, the temperature of the furnace was
decreased down to 300 °C and maintained constant for 120 h, in order to measure the radon
change;

- EXP4. The thermally stressed rock sample obtained from EXP3 was disaggregated along the
main macroscopic fault planes caused by the extremely fast heating/cooling cycles. Seven
large rock fragments with centimetric dimensions were recovered. All the rock fragments
were placed in the furnace preheated at 300 °C and the radon signal was monitored for 120 h.

24 132 2.3. Radon correction method

Radon gas emitted from rock samples was measured through a radon monitor (RAD7,
DurrIDGE Company) connected to the furnace in a closed-loop configuration and equipped with a
solid state silicon alpha detector (see [28] for further details). The activity concentration of thoron
was measured in continuum during experiments with an acquisition time of 1 h for each single
measurement (Table 1S). According to Tuccimei *et al.* [37], laboratory investigations are greatly
facilitated by the short half-life (56 seconds) of ^{220}Rn compared with the half-life (3.82 days) of
139 ^{222}Rn . The thoron atoms reach rapidly the equilibrium activity concentration and respond almost
140 instantaneously to any physicochemical change of the rock samples. Moreover, the geochemical
behaviour of ^{220}Rn is identical to that of ^{222}Rn , due to the lack of isotopic fractionation between
heavy radon and thoron isotopes with an extremely low mass difference of 0.01%.

To correctly interpret the experimental data, a decay (D) correction was applied to the
measured thoron signal ($^{220}\text{Rn}_M$), by considering the travelling time of atoms during gas transport
through the circuit and the effect of high absolute humidity (AH) onto the efficiency of the silicon
detector (Table 1S):

$$^{220}\text{Rn} = ^{220}\text{Rn}_M / D \times AH$$
 Eqn. (1)

Where D is equal to 0.1375, as the result of (i) the air flow rate through the circuit (0.8 L min^{-1}), (ii) the volume of the experimental circuit placed upstream of the radon monitor (2.53 L) and (iii) the ^{220}Rn decay constant (0.756 L min^{-1}). The mass water molecules hosted in the inner volume of RAD7 (gH_2O) may also neutralise part of the radon daughters (^{218}Po ions), lowering the efficiency of the silicon detector. The corrected value of AH (Table 1S) is derived by the expression [47]:

$$AH = -m \times gH_2O + c \quad \text{Eqn. (2)}$$

Where m and c are, respectively, the slope and the intercept of linear regression fit of efficiency data plotted vs. gH_2O [47].

2.4. Petrophysical analysis

Rock petrophysical properties and microstructural characterisation were performed at the HP-HT Laboratory of the INGV. Ultrasonic P- (V_p) and S- (V_s) wave velocities and porosity were measured on both intact and thermally stressed rocks by cutting three cubes (3 cm/side) of material from the from the upper, central and lower portion of each cylindrical sample, in order to calculate a representative average value (Table 2S). V_p and V_s were acquired using a high voltage (1000 V) pulse generator and a Tektronix DPO4032 oscilloscope and two piezoelectric transducer crystals (100 kHz to 1 MHz frequency). The value of porosity (ϕ) was determined using a helium pycnometer (AccuPyc II 1340) with $\pm 0.01\%$ accuracy (Table 2S).

The microstructural analysis was carried out by an optical petrographic microscope and a Jeol-JSM6500F Field Emission Gun Scanning Electron Microscope (FE-SEM) equipped with an energy-dispersive X-ray microanalysis system (EDS). High-magnification FE-SEM microphotographs were acquired in the backscattered electron (BSE) imaging mode, whereas low-

174 magnification optical photomicrographs were acquired under cross-polarised and parallel-polarised
light.

X-ray Powder Diffraction (XRPD) patterns were collected with a Siemens D5005
diffractometer operating in the θ - 2θ vertical configuration, equipped with a Ni-filtered $\text{CuK}\alpha$
radiation, installed in the Department of Ingegneria & Geologia of the University G. d'Annunzio.
Each XRPD spectra were recorded between 4 and 80 ° of 2θ , with a step scan of 0.02 ° and a
counting time of 8 s (Table 3S). XRPD patterns were first qualitatively evaluated to identify
crystalline phases by a search-match comparison with the commercial Inorganic Crystal Structure
Database (ICSD). The crystalline phases that better match XRPD patterns, i.e., better reproduce
positions and intensity of observed Bragg reflections, were then used as input crystallographic data.
The identification of crystalline phases for each pattern was performed using crystal models
allowing a better fit to the observed Bragg reflection (2θ positions and relative intensities).

**3. Thermally-induced rock physicochemical changes**

From a mineralogical point of view, the thermal treatment conducted from 100 to 900 °C did
not substantially change the phase assemblage of the rock either at the phenocryst or the matrix
scale. The comparison between XRPD spectra obtained from the intact rock, EXP1 and EXP3
shows that the position and intensity of the mineral peaks are always equivalent (Fig. 1). Moreover,
XRPD data do not reveal the appearance or disappearance of any crystalline phase during radon
thermal experiments (Fig. 1), in agreement with the high thermal stability of the anhydrous igneous
paragenesis [48-49]. Note that, at ambient pressure and under anhydrous conditions (such as those
from this study), leucite, clinopyroxene and magnetite melt congruently at temperatures (1,391-
1,686 °C; [50-51]) much higher than those investigated in this study.

The intact rock sample is massive in hand specimen (Fig. 2a) and no millimetre-sized
fractures are visible using the optical microscopy (Fig. 2b). The phonolite is composed of a dense
polymineralic aggregate with no apparent discontinuities. The only exception is represented by

several microcracking phenomena observable by BSE imaging at the micrometric to
submicrometric scale (Fig. 2b). These small inter- and intra-crystal microcracks are typical of
natural effusive products due to the effect of heat and gas loss during lava flow and emplacement.
The value (3.6%) of ϕ is relatively low (Table 2S), whereas V_p (5.51 km s⁻¹) and V_s (3.84 km s⁻¹) are
in the range documented for massive lava flow formations [21,38]. On the other hand, EXP1 and
EXP2 exhibit textural and petrophysical features different from those observed for the intact sample.
The applied heating and cooling rates (i.e., $\Delta T/\Delta t \gg 1$ °C/min) are fast enough to induce significant
thermal gradients through the rock sample [46,52]. This thermal stress condition leads to crack
opening accompanied by an increase in ϕ and a decrease in V_p and V_s , as generally reported for
highly crystalline rocks [46,52-55]. Several fractures are visible by eye in hand specimen (Fig. 2c),
as well as an almost continuous pattern of microcracks is observable at the micrometric scale (Fig.
2d). Within the analytical uncertainty, the petrophysical properties measured for EXP1 and EXP2
are almost identical, corresponding to 5.2 and 5.4% ϕ , 5.38 and 5.42 km s⁻¹ V_p and 3.63 and 3.68 km
s⁻¹ V_s , respectively (Table 2S). The comparable textural and petrophysical features indicate that a
similar level of microcrack damage was imparted on the rock sample during EXP1 and EXP2,
either by using temperature steps (100 °C ΔT) up to 900 °C or by applying a restricted number of
heating/cooling cycles (300 and 600 °C ΔT). In this context, mineral thermodynamic data outline
that leucite, clinopyroxene and magnetite are subjected to only limited thermal expansions at
subvolcanic temperatures [56]. Moreover, a low number of heating/cooling cycles may not enhance
significantly the natural crack patterns inherited at the time of eruption during lava flowing and
cooling onto the surface [21,53].

In contrast, in EXP3 and EXP4, the rock sample is exposed to a more efficient thermal stress
regime attained by applying twenty heating/cooling cycles and a strong thermal gradient (800 °C
ΔT). Millimeter-sized macroscopic faults progressively develop on the sample surface with
increasing the number of heating/cooling cycles (Fig. 3a). At the end of the thermal treatment, the

phonolite is disaggregated in seven large rock fragments along the main fault planes (Fig. 3b) and a
pervasive network of millimetric to micrometric cracks characterises the rock sample (Fig. 3c). Due
to the thermal stress and the resulting microfracture network, the value of ϕ increases up to 8.2%
(Table 2S). Conversely, V_p and V_s decrease down to 5.22 km s⁻¹ and 3.41 km s⁻¹, respectively, as a
proxy for the enhanced thermal damage (Table 2S). Therefore, both the high number of
heating/cooling cycles and the strong thermal gradient cause a rapid thermal shock. The intense
microfracturing is the consequence of the thermal stress condition imparted by expansion or
contraction of the rock grains up to levels corresponding to the tensile or shear strength of the
phonolite rock [54,57].

4. Changes in radon signal during thermal experiments

Due to the stochastic process of radioactive decay, the ²²⁰Rn vs. t diagram (Fig. 4a) shows
that, for each constant temperature, the radon signal may deviate from the average value within a
range of ~20-50%, responding to the statistical fluctuations of ²²⁰Rn measurements over the
annealing time. Since the recoil effect is practically T -independent [58], the role of temperature is to
increase the number of effective collision of radon atoms with other molecules and rock grain
boundaries, thus favouring the migration of radon gas through the microcracks of the phonolite
[36,37]. Coherently, EXP1 shows that ²²⁰Rn monotonically increases of about two orders of
magnitude as the temperature increases from 100 to 900 °C (Fig. 4b). This behaviour corroborates
the strong dependence of radon diffusion on temperature, leading to transient radon emissions due
to a new value of equilibrium activity concentration [28,58,59]. The change in radon emission rate
(E in arbitrary units) is sympathetically correlated with the temperature-dependent diffusion
coefficient of radon by a non-linear Arrhenius equation (where T is expressed in Kelvin; [60-62].
From an empirical point of view, the plot $\ln E$ vs. $1/T$ (Fig. 4c) provides a quantitative description of
the temperature-dependent migration of radon atoms through the microcracks of the rock sample
(see [28] for a detailed mathematical treatment). The regression fit of experimental data from this

study (Fig. 4c) yields a high correlation coefficient ($R^2 = 0.97$), attesting the close relationship
between the enhancing effect of temperature and the escape of radon atoms from the interconnected
fractures of the rock sample.

Results from EXP2 confirm that ^{220}Rn is almost instantaneously controlled by temperature
changes, showing either increasing or decreasing values as a function of the heating/cooling cycles
applied to the phonolite (Fig. 5a). The measurement of a transient radon signal confirms the close
relationship between temperature and the diffusion of radionuclides through the rock structural
irregularities which serve as radon diffusion paths [58,61]. Importantly, the value of ^{220}Rn
monitored for each single low-temperature (300 °C ΔT) and high-temperature (600 °C ΔT) cycle is
comparable to that recorded during EXP1 (Fig. 4a), despite the different thermal treatments adopted.
Moreover, the background level of radon recorded at the end of the heating/cooling cycles of EXP2
is constrained in a restricted range (2,664-3,791 Bq m⁻³; Fig. 5b), suggesting that the amount of
microcrack damage imparted by the thermal gradients is not sufficient to cause detectable ^{220}Rn
modifications. According to Mollo *et al.* [38], no significant radon changes are recorded in
laboratory during the deformation of low porosity and high strength crystalline phonolite rocks.
Most of the radon atoms remain entrapped within the inter- and intra-granular microcracks, until the
formation of macroscopic faults acting as new exhalation surfaces for the radon gas [63]. The
limited microcrack pattern developed in the thermally stressed phonolite (Fig. 2c-d) does not
sensibly increase the number of diffusion paths serving for the gas release. In fact, the evolution of
microfracture networks in volcanic rocks depends strongly on the efficacy of temperature changes
in opening or not new degassing paths [46]. The comparison between Fig. 4b and Fig. 5b indicates
that the radon signal from EXP2 is primarily controlled by the temperature dependence of the radon
emission rate, thus resembling the values of ^{220}Rn measured at 300, 600, and 900 °C during EXP1.

In the EXP3, the thermal gradient condition caused by twenty heating/cooling cycles with
800 °C ΔT is strong enough to increase substantially the number of diffusion paths and/or surface
area, by developing a pervasive crack network (Fig. 3). The ^{220}Rn vs. t diagram (Fig. 6a) shows that

the radon signal monitored for 120 h at 300 °C is remarkably higher than the temperature dependence of radon transport measured at 300 °C during EXP1 and EXP2. The release of gas trapped into the microstructural fabric of the rock is constant over time (Fig. 6a), denoting the attainment of a steady-state condition corresponding to a stationary radon emission [37]. The background level of radon (5,891-7,629 Bq m⁻³; Fig. 6b) measured during EXP3 is increased by ~110% with respect to that monitored during EXP2 (Fig. 5b). The intense rock fracturing enhances the number of diffusion paths [42], attesting that the level of microcrack damage imparted on the phonolite is sufficient to drive changes in radon emanation [64]. According to Sengupta *et al.* [65], the concentration of ²²⁰Rn radionuclides in a rock is proportional to its microstructural fabric where microfractures act as pathways for migration of radon atoms. The fractures serve as channels to connect the isolated microcracks to the crystalline rock network. These migration channels develop in the rock structure and connect the isolated microcracks to the external rock surface, so that radon liberated through interconnected fractures can easily migrate in the monitoring system [63].

The thermal shock condition attained in EXP3 (i.e., rapid expansion/contraction of the rock grains) causes sample rupture by formation of seven large rock fragments along the principal millimetre-sized macroscopic faults (Fig. 3b). The radon signal emitted from these rock fragments is measured during EXP4 for 120 h at the constant temperature of 300 °C (Fig. 7a). The presence of macroscopic rupture surfaces intensifies dramatically the radon emission, due to an enhanced migration of ²²⁰Rn atoms that escape more efficiently from the rock fragments [36,37]. The stochastic radioactive decay is accompanied by strong statistical fluctuations of ²²⁰Rn emissions, with deviations from the average value in the range of ~50-150%. Moreover, the background level of radon (5,885-19,887 Bq m⁻³; Fig. 7b) increases up to ~420% with respect to that measured in EXP2 for the temperature dependence of the radon emission rate. Therefore, it is concluded that the main effect of thermal shock on the highly crystalline phonolite is that crack growth develops rapidly into macrofractures, thus forming macroscopic faults [54,66,67]. The enlarged surface area of the phonolite sample (Fig. 3b) results in a new value of equilibrium activity concentration of

303 radon and the increased fraction of accessible ^{220}Rn atoms scales with the rock exhalation surface
[38,64,65]. Since the radon signal is proportional to the exhaling surface area, phonolite samples
with relative small matrix minerals may also expose high grain boundary surfaces, thus enhancing
remarkably the radon emission rate [63].

**5. Conclusive remarks**

We have investigated the effect of temperature on the radon signal emitted from a highly
crystalline phonolitic rock exposed to variable thermal conditions in the range of 100-900 °C. Our
experimental data indicate that a transient radon signal is produced by the temperature-dependent
diffusion of radionuclides through the rock structural irregularities which serve as preferential gas
pathways. However, this condition is attained only when thermal gradients do not induce substantial
modifications in the rock structure and relatively low number of new cracks is introduced by the
increasing temperature. In contrast, intense heating/cooling cycles can dramatically alter the
petrophysical properties of the rock, by generating microfractures through thermal expansion and
contraction of minerals. The diffusive microfracturing and the formation of macroscopic faults
produce new exhalation surfaces  that enhance the radon signal up to a stationary level that is much
higher than the temperature-dependent transient changes.

**Data accessibility.** The experimental data of this article have been uploaded as electronic
supplementary materials.

**Authors' contributions.** S.M. and P.T. designed and performed the radon experiments. G.G.
performed the experiments and applied correction to the radon signal. G.I. performed the
mineralogical and petrographical analyses. M.S. and P.S. contributed to writing the manuscript. All
the authors give final approval for publication.

**Competing interests.** The authors declare no competing interests.

**Acknowledgements.** Authors would like to thank Martina Mattina for her the help provided during
laboratory activities.

**References**

- 1. Connor CB, Clement BM, Song XD, Lane SB, West-Thomas J. 1993 Continuous
monitoring of high-temperature fumaroles on an active lava dome. Volcano Colima,
Mexico: Evidence of mass flow variation in response of atmospheric forcing. *J. Geophys.*
*Res.* **98 (B11)**, 19713-19722.
- 2. Monnin MM, Seidel JL. 1997 Physical models related to radon emission in connection with
dynamic manifestations in the upper terrestrial crust: a review. *Rad. Meas.* **28**, 703–712.
- 3. Alparone S, Andronico D, Giammanco S, Lodato L. 2004 A multidisciplinary approach to
detect active pathways for magma migration and eruption at Mt. Etna (Sicily, Italy) before
the 2001 and 2002–2003 eruptions. *J. Volcanol. Geotherm. Res.* **136**, 121–140.
- 4. Segovia N, Armienta MA, Valdes C, Mena M, Seidel JL, Monnin M, Pena P, Lopez MBE,
Reyes AV. 2003 Volcanic monitoring for radon and chemical species in the soil and in
spring water samples. *Rad. Meas.* **36**, 379–383.
- 5. Immè G., La Delfa S, Lo Nigro S, Morelli D, Patanè, G. 2006 Soil radon concentration and
volcanic activity of Mt. Etna before and after the 2002 eruption. *Rad. Meas.* **41**, 241–245.
- 6. Morelli D, Immè G, La Delfa S, Lo Nigro S, Patanè G. 2006 Evidence of soil radon as tracer
of magma uprising at Mt. Etna. *Rad. Meas.* **41**, 721–725.
- 7. Giammanco S, Sims KWW, Neri M. 2007 Measurements of ²²⁰Rn and ²²²Rn and CO₂
emissions in soil and fumarole gases on Mt. Etna volcano (Italy): implications for gas
transport and shallow ground fracture. *Geochem. Geophys. Geosy.* **8**, Q10001.
- 8. La Delfa S, Immè G, Lo Nigro S, Morelli D, Patanè G, Vizzini F. 2007 Radon
measurements in the SE and NE flank of Mt. Etna (Italy). *Rad. Meas.* **42**, 1404–1408.

9. Cox ME. 1983 Summit outgassing as indicated by radon, mercury and PH mapping, Kilauea
volcano, Hawai. *J. Volcanol. Geotherm. Res.* **16**, 131–151.
10. Cox ME, Cuff EK, Thomas MD 1980 Variations of ground radon concentrations with
activity of Kilauea volcano, Hawaii. *Nature* **288**, 74–76.
11. Heiligamann M, Stix J, Williams-Jones G, Sherwood Lollar B, Garzon GV. 1997 Distal
degassing of radon and carbon dioxide on Galeras volcano, Colombia. *J. Volcanol.*
*Geotherm. Res.* **77**, 267–283.
12. Dobrovolsky IP, Zubkov SI, Miachkin VI. 1979 Estimation of the size of earthquake
preparation zones. *Pure Appl. Geophys.* **117**, 1025–1044.
13. Fleischer RL. 1981 Dislocation model for radon response to distant earthquakes. *Geophys.*
*Res. Lett.* **8**, 477–480.
14. Thomas DM, Cox ME, Cuff KE. 1986 The association between ground gas radon variations
and geologic activity in Hawaii. *J. Geophys. Res.* **91**, 2186–12198.
15. Finizola A, Ricci T, Deiana R, Barde Cabusson S, Rossi M, Praticelli N, Giocoli A,
Romano G, Delcher E, Suski B, Revil A, Menny P, Di Gangi F, Letort J, Peltier A,
Villasante-Marcos V, Douillet G, Avaré G, Lelli M. 2010 Adventive hydrothermal
circulation on Stromboli volcano (Aeolian Islands, Italy) revealed by geophysical and
geochemical approaches: implications for general fluid flow models on volcanoes. *J. Volc.*
*Geotherm. Res.* **196**, 111–119.
16. Ricci T, Finizola A, Barde-Cabusson S, Delcher E, Alparone S, Gambino S, Miluzzo V.
2015 Hydrothermal fluid flow disruptions evidenced by subsurface changes in heat transfer
modality: the La Fossa cone of Vulcano (Italy) case study. *Geology* **43**, 959–962.
17. Gasparini P., Mantovani MSM. 1978 Radon anomalies and volcanic eruptions. *J. Volcanol.*
*Geotherm. Res.* **3**, 325–341.

[revised manuscript text omitted]

- 53. Vinciguerra S, Trovato C, Meredith PG, Benson PM. 2005 Relating seismic velocities,
thermal cracking and permeability in Mt. Etna and Iceland basalts. *Int. J. Rock Mech. Min.
Sci.* **42**, 900–910.
- 54. Browning J, Meredith P, Gudmundsson A. 2016 Cooling-dominated cracking in thermally
stressed volcanic rocks. *Geophys. Res. Lett.* **43**, 8417–8425.
- 55. Zhang F, Zhao J, Hu D, Skoczylas F, Shao J. 2018 Laboratory Investigation on Physical and
Mechanical Properties of Granite After Heating and Water-Cooling Treatment. *Rock Mech.
Rock Eng.* **51**, 677–694.
- 56. Ghiorso MMS, Sack ROR. 1995 Chemical mass transfer in magmatic processes IV.
A revised and internally consistent thermodynamic model for the interpolation and
extrapolation of liquid–solid. *Contrib. Min. Petr.* **119**, 197–212.
- 57. David C, Menéndez B, Darot M. 1999 Influence of stress-induced and thermal cracking on
physical properties and microstructure of La Peyratte granite. *Int. J. Rock Mech. Min Sci.* **36**,
433–448.
- 58. Beckman IN, Balek V. 2002 Theory of emanation thermal analysis XI. Radon diffusion as
the probe of microstructure changes in solids. *J. Therm. Anal. Calorim.* **67**, 49–61.
- 59. Voltaggio A, Masi U, Spadoni M, Zampetti G. 2006 A methodology for assessing the
maximum expected radon flux from soils in Northern Latium (Central Italy). *Environ.
Geochem. Health* **28**, 541–551.
- 60. Glasstone S, Laidler KJ, Eyring H. 1941 *The Theory of Rate Processes: The Kinetics of
Chemical Reactions, Viscosity, Diffusion and Electrochemical Phenomena*, p. 611,
McGraw-Hill, New York.
- 61. Balek V, Beckman IN. 2005 Theory of emanation thermal analysis XII. Modelling of radon
diffusion release from disordered solids on heating. *J. Therm. Anal. Calorim.* **82**, 755–759.
- 62. Shewmon PG. 2016 *Diffusion in Solids*. The Minerals, Metals & Materials Series. Springer
International Publishing, pp. 241. (doi: 10.1007/978-3-319-48206-4).

63. Mollo S, Tuccimei P, Soligo M, Galli G, Scarlato P. 2018 *Advancements in understanding*
*the radon signal in volcanic areas: A laboratory approach based on rock physico-chemical*
*changes*. In *Integrating Disaster Science and Management*. Editors: Pijush Samui, Dookie
Kim, and Chandan Ghosh. Publisher: Elsevier. pp. 309-328. ([https://doi.org/10.1016/B978-](https://doi.org/10.1016/B978-0-12-812056-9.00018-X)
[0-12-812056-9.00018-X](https://doi.org/10.1016/B978-0-12-812056-9.00018-X)).
64. Banerjee KS, Basu A, Guin R, Sengupta D. 2011 Radon (^{222}Rn) level variations on a
regional scale from the Singhbhum Shear Zone, India: a comparative evaluation between
510 influence of basement U-activity and porosity. *Radiat. Phys. Chem.* **80**, 614–619.
65. Sengupta D., Ghosh A., Mamtani MA. 2005 Radioactivity studies along fracture zones in
areas around Galudih, East Singhbhum, Jharkhand, India. *Appl. Rad. Isot.* **63**, 409–414.
66. Brace WF, Paulding Jr BW, Scholz CH. 1966 Dilatancy in the fracture of crystalline rocks.
*J. Geophys. Res.* **71**, 3939–3953.
67. Wawersik WR. 1972 *Time-dependent rock behavior in uniaxial compression*, in:
*Proceedings of 14th Symp. Rock Mech.*, Penn. State Univ., University Park, Pa, USA, 85–
106.

**Figure captions**

Figure 1. X-ray powder diffraction (XRPD) spectra tracking the stability of minerals in the intact
phonolitic rock and during thermal experiments (EXP1 and EXP3).

Figure 2. Representative structural and textural features of the intact rock sample (a) and
experimental sample from EXP1 and EXP2 (b). The cylindrical rock sample is 50 mm in diameter
and 110 mm in length. Microphotographs obtained by optical microscopy and FE-SEM are acquired
at the millimetric and micrometric scale, respectively. The arrows indicates both microcracks and
macroscopic fractures. Aug, augite. Lct, leucite. Mag, magnetite.

Figure 3. Representative structural and textural features of the rock sample from EXP3 and EXP4
obtained during intense heating/cooling cycles (a), at the end of thermal stress experiment (b), and
at the millimetric and micrometric scale by optical microscopy and FE-SEM, respectively. The
arrows indicates both microcracks and macroscopic fractures. Cpx, clinopyroxene. Lct, leucite. Mag,
magnetite.

Figure 4. Results from EXP1 plotted in ^{220}Rn vs. t (a), ^{220}Rn vs. T (b) and $E_{220\text{Rn}}$ vs. $10,000/T$ (c)
diagrams. The rock sample was first kept at 100 °C for 48 h. Then, through a heating ramp (Δt) of 5
538 min, the temperature (ΔT) was increased by 100 °C and maintained constant for 22 h. This
incremental step was replicated eight times (5 min Δt and 100 °C ΔT) up to the maximum
experimental temperature of 900 °C. See Mollo *et al.* [28] for a detailed mathematical treatment of
the non-linear Arrhenius equation describing the dependence of radon emission rate on temperature.

Figure 5. Results from EXP2 plotted in ^{220}Rn vs. t (a) and ^{220}Rn vs. T (b) diagrams. The rock
sample was first kept at 300 °C for 48 h. Then, the rock sample was subjected to low- and high-
thermal stress cycles. For the low-temperature stress cycles (300 °C ΔT), the rock sample was
heated from 300 °C to 600 °C (5 min Δt) and, after 1 h, was cooled down to 300 °C at the same rate.
At the end of third cycle, the temperature was maintained constant at 300 °C for 48 h. The same
strategy was applied for the high-temperature thermal stress cycles (600 °C ΔT) by increasing the
temperature from 300 to 900 °C.

Figure 6. Results from EXP3 plotted in ^{220}Rn vs. t (a) and ^{220}Rn vs. T (b) diagrams. More intense
heating/cooling cycles were performed with the aim to thermally stress the rock sample and to
induce microfracturing. The rock sample was heated from 100 to 900 °C (5 min Δt) and then was
cooled from 900 to 100 °C at the same rate. This heating/cooling cycle (5 min Δt and 800 °C ΔT)

was repeated twenty times. Due to the short acquisition time, the radon signal was not monitored
during the fast heating/cooling steps. However, at the end of the twentieth cycle, the temperature of
the furnace was decreased down to 300 °C and maintained constant for 120 h, in order to measure
the radon change.

Figure 7. Results from EXP4 plotted in ^{220}Rn vs. t (a) and ^{220}Rn vs. T (b) diagrams. The thermally
stressed rock sample obtained from EXP3 was disaggregated along the main macroscopic fault
planes caused by the extremely fast heating/cooling cycles. Seven large rock fragments with
centimetric dimensions were recovered. All the rock fragments were placed in the furnace preheated
at 300 °C and then the radon signal was monitored for 120 h.

X-ray powder diffraction (XRPD) spectra tracking the stability of minerals in the intact phonolitic rock and during thermal experiments (EXP1 and EXP3).

416x425mm (72 x 72 DPI)

Intact rock sample, EXP1 and EXP2

Representative structural and textural features of the intact rock sample (a) and experimental sample from EXP1 and EXP2 (b). The cylindrical rock sample is 50 mm in diameter and 110 mm in length.

Microphotographs obtained by optical microscopy and FE-SEM are acquired at the millimetric and micrometric scale, respectively. The arrows indicate both microcracks and macroscopic fractures. Aug, augite. Lct, leucite. Mag, magnetite.

499x493mm (72 x 72 DPI)

EXP3 and EXP4

Representative structural and textural features of the rock sample from EXP3 and EXP4 obtained during intense heating/cooling cycles (a), at the end of thermal stress experiment (b), and at the millimetric and micrometric scale by optical microscopy and FE-SEM, respectively. The arrows indicates both microcracks and macroscopic fractures. Cpx, clinopyroxene. Lct, leucite. Mag, magnetite.

333x618mm (72 x 72 DPI)

Results from EXP1 plotted in ²²⁰Rn vs. t (a), ²²⁰Rn vs. T (b) and E_{220Rn} vs. 10,000/T (c) diagrams. The rock sample was first kept at 100 °C for 48 h. Then, through a heating ramp (□t) of 5 min, the temperature (□T) was increased by 100 °C and maintained constant for 22 h. This incremental step was replicated eight times (5 min □t and 100 °C □T) up to the maximum experimental temperature of 900 °C. See Mollo et al. [28] for a detailed mathematical treatment of the non-linear Arrhenius equation describing the dependence of radon emission rate on temperature.

416x625mm (72 x 72 DPI)

Results from EXP2 plotted in ^{220}Rn vs. t (a) and ^{220}Rn vs. T (b) diagrams. The rock sample was first kept at 300 $^\circ\text{C}$ for 48 h. Then, the rock sample was subjected to low- and high-thermal stress cycles. For the low-temperature stress cycles (300 $^\circ\text{C}$ \square T), the rock sample was heated from 300 $^\circ\text{C}$ to 600 $^\circ\text{C}$ (5 min \square t) and, after 1 h, was cooled down to 300 $^\circ\text{C}$ at the same rate. At the end of third cycle, the temperature was maintained constant at 300 $^\circ\text{C}$ for 48 h. The same strategy was applied for the high-temperature thermal stress cycles (600 $^\circ\text{C}$ \square T) by increasing the temperature from 300 to 900 $^\circ\text{C}$.

499x498mm (72 x 72 DPI)

Results from EXP3 plotted in ^{220}Rn vs. t (a) and ^{220}Rn vs. T (b) diagrams. More intense heating/cooling cycles were performed with the aim to thermally stress the rock sample and to induce microfracturing. The rock sample was heated from 100 to 900 °C (5 min \square t) and then was cooled from 900 to 100 °C at the same rate. This heating/cooling cycle (5 min \square t and 800 °C \square T) was repeated twenty times. Due to the short acquisition time, the radon signal was not monitored during the fast heating/cooling steps. However, at the end of the twentieth cycle, the temperature of the furnace was decreased down to 300 °C and maintained constant for 120 h, in order to measure the radon change.

499x498mm (72 x 72 DPI)

Results from EXP4 plotted in ^{220}Rn vs. t (a) and ^{220}Rn vs. T (b) diagrams. The thermally stressed rock sample obtained from EXP3 was disaggregated along the main macroscopic fault planes caused by the extremely fast heating/cooling cycles. Seven large rock fragments with centimetric dimensions were recovered. All the rock fragments were placed in the furnace preheated at 300 °C and then the radon signal was monitored for 120 h.

499x498mm (72 x 72 DPI)

Appendix D

Associate Editor's comments (Professor Axel Schmitt):

Dear Dr. Mollo,

Your manuscript has now been reviewed by three experts in the field. As you can see, a good measure of criticism has been voiced, and although one of the reviewers recommends rejection, I feel that even this most critical review recognizes the potential of your research. However, there is clearly a need for substantial changes before acceptance can be considered.

For making these changes, you will find many helpful comments in the reviews, and I concur with most of them. I would like to reinforce that the current title is misleading: most petrologists will not associate the rock that you used in your experiments with the qualifier "highly crystalline" as the title suggests. A more specific and adequate title is therefore required.

The manuscript has many deficiencies in the use of standard English. In addition to grammatical and semantic errors pointed out by the reviewers, there are often awkward and unnecessary complicated phrases. For example in the abstract, please consider potential corrections for:

“thin-to-thick” (of variable thickness)

“lithological formations” (rocks)

“physicochemical features” (physical and chemical properties)

“In this respect” (To assess radon mobilization in subvolcanic thermal regimes, we have performed ...)

“However, it has been also observed that” (not needed)

“produces” (produced)

“intensifies dramatically” (reverse, as suggested by reviewer, and state “intensified”, as you are still describing your experiments)

“giving reason for” (which explains that ...)

There should be a much clearer concluding sentence at the end of the abstract what the implications of your research are, especially addressing a broader audience.

Regards,

Axel Schmitt (Heidelberg University)

Dear Editor,

We are very grateful to the three anonymous Reviewers for their efforts and the work done on the original manuscript. We stress that their comments have been seriously considered during the review process. Thereby, the title, abstract and the overall text have been improved and carefully revised following the suggestions of the Associate Editor (blue text), Reviewer#1 (red text), Reviewer#2 (violet text) and Reviewer#3 (green text).

We truly acknowledge the efforts done by the Reviewer#1 but, sadly, we were forced to decline about half of his/her suggestions. We recognize that he/she is an expert in the field, but we do not think that his/her expertise matches with our main topic, especially for the measurement of radon emissions from natural (microfractured and heterogeneous) rocks using an alpha-counting system. For our reference scientific community, this analytical method is the most common and accurate for the measurement of radon emissions in active tectonic and volcanic areas. Although it is not our intention to disappoint or upset anybody, we are sorry to say that most of the comments of the Reviewer#1 allude to aspects (i.e., starting materials, experimental methods, analytical methods and references) very different from those discussed in our work, thus resulting inappropriate. This basically explains the contrasting point of view of the Reviewer#1 relative to those expressed by the

Reviewer#2 and Reviewer#3. More details are provided in the text below and we hope that you will impartially consider our responses, especially because we closely agree with all of the comments of the Reviewer#2 and Reviewer#3, now totally addressed in the revised manuscript.

Sincerely,
Silvio Mollo and co-authors

Reviewer: 1

This article reports the gaseous measurement of the activity concentration of thoron (radon-220) released from a phonolitic rock subjected to heating events and heating/cooling cycles from 100°C to 900°C, characteristic temperatures that may result from magmatic intrusion underneath a volcanic edifice. A total of four experiments performed show that transient increases in thoron release occur during heating. The authors argue that temperature-dependent diffusion controls the release of thoron and that intense heating/cooling cycles form micro-fractures which interconnect, hence opening new pathways for thoron release.

The paper is generally easy to read and to follow. The figures also give a clear message to the reader. The general organization of the paper is acceptable.

We thanks the Reviewer#1 for this comment.

However, the points detailed below (major comments) deserve particular attention:

(1) The title and the introduction section are misleading as they only present radon, which generally corresponds to radon-222 isotope, and not to thoron (radon-220 isotope) which is measured in this study. I would suggest correcting this throughout the manuscript and in the title.

This comment is not entirely correct. Radon is a chemical element of the periodic table and corresponds to 3 different isotopes, i.e., ^{222}Rn , ^{220}Rn and ^{219}Rn . We agree that the name of the isotope can be added to the title and introduction just to provide an additional information.

In addition, one volcanic rock type is studied only, not several “highly crystalline rocks”.

We totally agree with this comment. The title has been modified accordingly.

Finally, about 20% of cited articles in the reference list are self-citations which may appear a little too much.

This comment is quite provocative, especially because our Research Team comprises a group of different scientists that, since 2006, are designing and calibrating novel methods and instrumentations for radon measurements which are periodically published in scientific journals. Should we be ashamed of this?

(2) The introduction section is rather short and not quantitative enough. Several works have studied the effects of temperature and pressure on radon (less on thoron) on various lithologies, but a little only is recalled in the introduction section, although almost all these works are cited. I would suggest adding a paragraph relying on the known/identified/suspected effects of temperature on the release of radon and thoron from porous materials such as rocks in the laboratory, giving quantitative numbers. Below, I inserted some references that could be used in this new paragraph.

We regret to reply that this is basically wrong and cannot be applied to our work. The papers suggested by the Reviewer#1 do not deal with lithologies and/or rocks but, rather, with crushed single minerals or powdered materials of non-igneous origin. In some cases, these papers do not present experimental data but computer simulations and/or measurement conducted at very low temperatures (i.e., maximum 40 °C) that are far from subvolcanic conditions. The analytical methods (Lucas cell and gamma-ray spectrometry) are also very different to that used in our work. Therefore, the Reviewer#1 is asking to write a new paragraph in which we should discuss about

materials, processes, thermal conditions and analytical methods never presented and discussed in the present study. Why should we accept a such inappropriate suggestion?

(3) The authors should differentiate transient release from steady-state release of thoron. In order to do that on a clearer manner, I would suggest to measure the thoron release from the samples before any heating experiment is performed and then to compare it with post-heating data. Not merely comparing equilibrium activity concentrations of thoron in the closed system, but instead to do that quantitatively, I would suggest calculating an effective emanation of thoron (or an effective radium-224 concentration), as it has been done routinely for radon (effective radium-226 concentration). This will give a physical value for the thoron emanating power of the studied samples. For example, the data plotted in Fig. 7, which gives thoron activity concentration at an infinite time compared with thoron half-life, should be used to obtain a thoron emanating power of the sample. This more quantitative approach will also give information on the reversibility or irreversibility of the (micro-) fracturing induced by the heating events and heating/cooling cycles.

The approach proposed by Reviewer#1 is very far from the experimental and analytical data presented in our study and generally adopted by the geochemical community investigating (and monitoring) active tectonic and volcanic areas. First, the used analytical system does not measure radium concentration. Second, the system is not appropriate for measuring the emanating power of rocks that is generally done by crushing the material to a certain grain size and by analyzing it with a gamma-ray counting system or a Lucas scintillation cell. Evidently, the Reviewer#1 is confusing different methods and approaches. It remains the fact that we cannot explain in a scientific article the basic principles of a generalised analytical technique (e.g., SEM, EPMA, XRPD, etc.). Third, the quantitative information requested by the Reviewer#1 on the reversibility or irreversibility of the (micro-)fracturing cannot be addressed by radon analysis. Rock fracturing is a physical property quantified with other and more appropriate techniques. We have presented porosity and P-S-wave velocity data, together with a detailed textural analysis. As documented in a vast literature, these three techniques are used to track the rock physical changes. Moreover, it is quite obvious that microcrack propagation and rock failure are irreversible processes at the scale of our experiments (i.e., a cm-sized rock). Indeed, the rock is fractured and/or disaggregated in several parts. Thus, the comment of Reviewer#1 on the reversibility or irreversibility of the process is difficult to understand.

Finally, we would like to bring to your attention some key information that can be found in the manual of the radon monitoring system commonly used by our community: “1) RAD7 is a Sniffer that detects the 3-minute alpha decay of a radon daughter, without interference from other radiations; 2) RAD7 is a true, real-time continuous monitoring system to analyse the variation of radon level during the period of the measurement; 3) for good data, it is important that there be sufficient counts to provide statistically precise readings; 4) devices which give just a single, average reading, or whose precision is inadequate except after a long measurement time, are not, in this sense, continuous monitors; 5) during radon measurements, the whole radon decay chain builds up inside the instrument, and the various daughters become well populated; 6) instruments that measure radon decay products in the air are called "working level" monitors. Working level monitors sample air through a fine filter and then analyze the filter for radioactivity. The radon progeny are metal and they stick to the filter and are counted by a working level instrument. Radon-222, an inert gas, passes through the filter, so it is not counted in such an instrument. Therefore, a working level instrument measures the radon progeny concentration (polonium-218, etc.), in the air, but not the radon gas concentration; 7) RAD7, on the other hand, measures radon gas concentration. Radon daughters do not have any effect on the measurement. RAD7 pulls samples of air through a fine inlet filter, which excludes the progeny, into a chamber for analysis. The radon in the RAD7 chamber decays, producing detectable alpha emitting progeny, particularly the polonium isotopes. The RAD7 detects progeny radiation internally, the only measurement it makes is of radon gas concentration. The two-sigma statistical uncertainty is then measured; 8) RAD7 uses a solid state

alpha detector. A solid state detector is a semiconductor material (usually silicon) that converts alpha radiation directly to an electrical signal. Its advantage is the ability to electronically determine the energy of each alpha particle. It makes possible to discriminate exactly which isotope (polonium-218, polonium-214, etc.) produced the radiation, so that you can immediately distinguish old radon from new radon, radon from thoron, and signal from noise”.

(4) The experiments carried out on the samples are worth of investigation. However, I would suggest associating every performed experiment to a natural phenomenon that may happen underneath a volcano, or elsewhere. I suspect that this point would necessitate new experiments, but this would deeply strengthen the scope of the paper.

We agree totally with this comment. More information were provided into the “Radon thermal experiments” section. Thank you!

(5) I am wondering how an apparatus dedicated to measure thoron activity concentration in the air under normal conditions is also able to measure it under high temperature (i.e. above normal operating temperature, which is generally below 45°C). At minimum, this will increase the noise level and probably the statistical dispersion of counts. Did you use any cooling procedure before the measurement in the apparatus? In addition, the sampling interval of one hour appears quite large compared with the thoron half-life of less than 1 min. For radon-222, a sampling interval of 10 min to 1 h is generally necessary to detect fast transient changes.

These comments are very confusing. Obviously, the radon detector is not exposed to temperatures up to 800 °C. We do not think that a such type of detector may exist. We reported in the original text that radon is bring to a RAD7 measuring system in a closed-loop configuration, which means air carrier transport. More details are now added to the text. However, it remains the fact that we wrote four pages (referring to Figure 1 and Table 1) about the temperature-dependence of radon diffusion from the rock pores and fractures (see also the cited references). This is the enhancing mechanism for the radon signal. Thus, point (5) is basically incorrect. The effect of temperature increases the mobility of atoms favouring their diffusion and increasing the radon emission from the rock sample. The sampling interval of 1 hour is a cumulative time (this has been now added to the revised text) and corresponds to the canonical RAD7 counting method for the alpha particles from the polonium decay. Since the number of atoms bring to the detector increases with temperature, the resulting counting statistics are also significantly improved (see also the cited references).

(6) The methodology does not include any uncertainty description. What are the sources of uncertainty and what is the experimental uncertainty for a given thoron activity concentration measurement?

This issue has been addressed by three different works of Tuccimei et al.. We cited only the most recent one, i.e., Tuccimei et al., 2015, in order to do not increase the number of self-citations (and differently to what stated above by the Reviewer#1). In the revised text, we have also added more details about the uncertainty description.

All these major points imply that, according to me, your paper cannot be accepted at this stage. I encourage you to carry out new experiments, making sure all the points mentioned above are checked, and to resubmit a stronger manuscript to Royal Society Open Science, or elsewhere. I hope that the comments above and below will help you in this task.

We disagree with this evaluation.

Other comments:

1) Line 49: Probably geochemical anomalies in active volcanic settings have not been observed in Italy only.

This comment is provocative. The further text includes a great number of citations from other volcanic settings.

2) Line 64: Please detail more what is the thermal weakening process and why the study of thoron should help.

This information is reported 7 lines below.

3) Line 72: There are many other papers describing this effect, including increases of porosity and permeability.

We know that there are hundreds of studies on this topic, but we must select only the literature most appropriate for our paper. What is the utility of this comment?

4) Lines 74-75: The effect on this “emanating power” should be explained in a dedicated paragraph. What are the general effects? Please try to be more quantitative.

We agree that the term “emanating power” is misleading in this sentence. The aim of our study is not to investigate a such effect, but rather the temperature-dependent diffusion of radon and its change under the effect of thermal microfracturing, i.e., the radon change.

5) Line 76: If we take time to look at what has been done, generally we have data, but the understanding of the processes involved is poorly known. Building a model reproducing data is more difficult. Proposing a simple model together with your data-set to explain it would be extremely rewarding.

We would be very happy to do this. But more experiments are necessary to correctly constrain the natural process, with particular regard for the role played by carries gases.

6) Line 92: “behaviour”.

This change has been now done.

7) Lines 103-105: This sentence should be moved to the introduction section.

This comment contrasts with that at the point 4)

8) Line 106: “under the following conditions”.

This change has been now done.

9) Lines 144-115: One hour may appear too much to measure thoron knowing its half-life of less than a minute.

We have now explained that the radon measure refers to a cumulative time typically used for radon survey in tectonic and volcanic areas.

10) Lines 123-124: I do not understand why it was not possible to measure thoron activity concentration continuously, while it has been done in the other experiments.

The reason is due to the lack of counting statistics. Do you know how the detector of RAD7 works?

11) Line 130: This stationary effect you are analyzing here should be studied using a steady-state physical quantity, such as the effective radium-224 concentration.

This aspect is beyond the aim of our work.

12) Line 134: Please add the manufacturer’s country.

This information has been now added.

13) Lines 137-139: One-hour sampling interval is also enough to measure radon-222 activity concentration.

This is wrong. In this case you would have very low counting statistics and a useless signal with a very high uncertainty!!!

14) Line 142: Please insert a reference to substantiate this assertion.

This has been now done.

15) Line 150: Is there any leakage of the system? Did you measure it using dedicated experiments?

The system is totally sealed and no leakage is observed. We published two papers about this.

16) Line 165: Please remove the extra words “from the”.

This has been now done.

17) Line 178: “installed at the”.

This has been now done.

18) Lines 184-185: This sentence is redundant with the sentences written just above.

We agree. The sentence has been deleted. Thank you.

19) Lines 188-190: This is counter-intuitive and does not match former studies showing mineral transformation(s) during heating experiments. Actually, we do see a difference in Fig. 1 when looking at the two largest peaks of the leucite between 25° and 30° (2θ), with a significant increase of the first peak after heating to 900°C. Please check.

The melting temperature of the minerals is much higher than 900 °C. The peak change occurs for all the mineral species and is due to the different intensity of the signal nor the mineral transformation or breakdown. This has been verified by Rietveld refinements.

20) Line 196: I agree with that, but partial melting can nevertheless appear, especially at 900°C.

This is not correct for a crystalline rock with phonolitic composition.

21) Lines 203 and 212: Please give uncertainties to appreciate the differences or similarities.

The uncertainties are listed in TableS2. They are 1-2 orders of magnitude lower than the measured value. For example, 3.6% of porosity has +/- 0.1% of error. This useless information would only makes the text more difficult to read.

22) Lines 245-246: How is your quantity E calculated? Generally E is used as the emanation factor for radon and thoron. You may use E as the emanation difference.

E is the relative change of Thoron and can be calculated from data in Table 1.

23) Line 246: Please do not use this adverb “sympathetically” here.

“Sympathetically” has been replaced with “positively”.

24) Lines 254-255: I agree with that point, but with the sampling time interval of 1 h, you cannot claim that the release of ²²⁰Rn is instantaneous after heating because you cannot measure it experimentally.

We totally agree. The word “instantaneous” has been deleted.

25) Lines 258-259: It appears here irreversible indeed. Please check whether all the heating procedures you used imply irreversible changes in your samples.

This is not true for EXP1 and EXP2. We have verified this effect several times.

26) Lines 269-270: Please rewrite this sentence.

This sentence has been modified according to the comment of Reviewer#2.

27) Lines 271-273: Please try to be more quantitative.

This sentence has been now done a few lines above according to the comment of Reviewer#2.

28) Line 277: What was the background value before the heating experiment? A given block can also show some significant heterogeneity between its core samples.

You cannot measure the radon signal of a large (tens of centimeters) block. It is obvious that the measurement is always limited to the size of the accumulation chamber that is on the order of some centimeters.

29) Lines 278-280: This last experiment is relatively obvious and should not be discussed this way. Please see the main comment above.

We disagree with this comment that is also in contrast with the suggestions of the Reviewer#2 and Reviewer#3.

30) Lines 287-289: This sentence is interesting, but does give any new message to the reader as this has already been observed.

It is not clear the direction of this comment. The sentence is interesting but unnecessary? What?

31) Lines 295-296: Why the statistical counting fluctuations are larger than in other experiments? Could it be the temperature of the gas entering the apparatus?

The statistical counting fluctuations just reflect the increased number of exhaling surface as reported in the text.

32) Line 302: Equilibrium activity concentration can be used to calculate a physical quantity.

This is not correct for our type of experiments in which you have heterogeneous natural rocks.

33) Line 304: You did not measure the exhaling surface area.

This is not correct for EXP3 and EXP4. The radon emission increases because the exhalation surface increases. There are not other possible mechanisms.

34) Lines 309-319: Generally, the conclusion section is different from the abstract. Please recall the main aspects of the paper and open some perspectives.

This has been now done.

35) Line 357: "Heiligmann". Line 486: Please correct the alignment. Please check the reference list.

This has been now done.

36) Fig. 1: Raw data are not so useful. Please quantify the changes.

We disagree with this comment. Fig. 1 is frequently used in analogous studies. See the figures reported in the cited references.

37) Figs. 2 and 3: Please rewrite the caption describing the subfigures a, b, c, and d separately.

We have used the style of the journal.

38) Fig. 4: The two steps of thoron concentration increase from 500 to 600°C and from 700 to 800°C are relatively larger than the others. Could you explain this observation? Is there any mineralogical transformation at these two temperatures?

This is the typical Arrhenian behaviour of a temperature-dependent diffusion equation, also in agreement with the regression fit showed in Fig. 4.

39) Line 547: “At the end of the third cycle”.

This has been now done.

40) Figs. 5, 6 and 7: The subplot b does not appear essential.

We disagree with this comment because the subplots show the radon increase with temperature that is the main focus of our work.

41) Figs. 6 and 7: This figure should be removed after the calculation of a physical quantity describing the thoron emanating power of the samples is done.

We disagree with this comment (see above).

42) Fig. 7: Were the samples entirely cooled during the measurement of thoron concentration?

This is in contrast with the aim of our study. The experimental system has been designed to analyse the signal in continuum during sample heating.

Reviewer: 2

Dear Editor,

I have read and reviewed “Transient to stationary radon emissions from highly crystalline rocks exposed to subvolcanic temperatures in laboratory” by Mollo et al. The paper described a set of experiments looking at the effect of temperature on rock radon exhalation. The results clearly show that temperature strongly controlled the rate of Rn production from whole rock cores, and that strong thermal cycling can cause enough thermal stress induced damage to increase the exhalation rate. I think this is an important result and worth of publication. The results appear to be real and the interpretation is scientifically sound. I have some minor editorial commentary designed to help the paper be clearer and more understandable, but recommend publication after these minor changes are made.

We are pleased that Reviewer#2 considers our results “important” and “worth of publication”, as well as “the interpretation is scientifically sound”. We have appreciated the comments of the Reviewer#2 either in this report or in the annotated pdf. The original manuscript has been modified accordingly.

Concerns:

There are two basic steps of Rn release from the rock core.

1) In step one the Rn must be released from the crystal lattice into the adjacent pore space and fracture network. This release can happen via alpha recoil during Rn production, mechanical fracture of mineral grains and subsequent release, and to a lesser extent diffusion through the mineral grains. This step is likely dominated by alpha recoil, which is not temperature dependent.

2) In the second step, the Rn moves through the pore and fracture network to the outside of the core. If there is no pressure gradient, this process is dominated by diffusion which is highly temperature dependent.

I am fairly certain the authors’ are mostly referring to process 2) when they discuss release results, but I think they should add even a few sentences to clarify the release process and which process they think is dominant.

Reviewer#2 is correct. The dominant process is that at point 2). We have now clarified this aspect.

I really think the authors should give at least a basic description of the analytical system rather than simply refer readers to another journal article. The article should stand by itself, and it is very hard to interpret laboratory results, without a basic understanding of the measurement apparatus.

We agree with this suggestion. The experimental system is now described in the text.

I think the conclusions could use some general advice for future researcher concerning when and where thermal cycling driven radon release will be an important process to concern. These should be order of magnitude quantitative, but give readers some quantitative advice on when this process should be considered important.

This point has been now introduced and discussed in the “Concluding remarks” section.

The English is decent, but could definitely use some tightening up. I have attached an annotated pdf with 27 additional comments and editorial suggestions.

Thanks for your kind work. We have greatly appreciated your efforts. The annotated pdf was used as a guide to improve the manuscript and the English text.

Reviewer: 3

A very interesting paper.

We are pleased for this very kind comment of Reviewer#3

A general wording comment: micrometric should be micrometer, etc throughout the paper.

This has been now done. Thank you.

The paper should be rewritten with improved sentence structure.

The text has been now improved.

What do you anticipate the effect of pressure from depth to have on the processes observed here at ambient pressure?

What we observe is a temperature-dependent diffusion of radon which is quite independent of pressure according to the canonical Arrhenius equation. Moreover, we have now explained that the limitation of radon emissions measured in laboratory is that they are not directly comparable to those monitored in natural systems. Our experiments may only show relative radon changes analogous to positive/negative anomalies typical of volcanic areas.

Please comment on the fine grained nature of the rock studied, in view grain size studies of thermal cracking; i.e. from surface area considerations one would not expect a great deal of thermal cracks from differential thermal expansion of mineral grains: how does this help or hinder the processes you are studying?

We closely agree with this comment. In the applied temperature range, igneous mineral phases undergo only a very limited thermal expansion. However, an important structural aspect of these crystals is that their isotropic thermal expansion changes remarkably as a function of the mineral species considered. As a consequence, the thermal expansion mismatch of the different mineral constituents instigates thermal microcracking by applying extremely fast heating/cooling cycles of with DELTA-time = 5 min and DELTA-Temperature = 800 °C. We have added this important concept in the revised text together with the related references.

60-reword

This has been now done.

62: redundant

This has been now changed according to the Reviewer#2

77: what does highly crystalline rock mean?

We have rephrased highly crystalline rock with phonolitic lava, following the suggestion of the Associate Editor.

92: what does characteristically brittle mean, please reference

It means microfracturing during deformation (Mollo et al., 2011). This information has been now added to the text.

102: I could not access reference [28] to read about the experimental setup, and we have had experience with many rock heating studies: a question arises as to how one can heat a sample of this size at the rates stated, with essentially no over run of temperature? please give enough detail in the paper to explain this--it ultimately may not matter given the results, but this reviewer would like to see this detail.

We have now reported a detailed description of the experimental setup, also following the comment of the Reviewer#2.

201:micrometer

This has been now done.

221- please explain "more efficient thermal stress"

This has been now replaced with greater temperature changes.

230-some believe that thermal shock is faster than the heating/cooling in these experiments

We have replaced "thermal shock" with "intense thermal microfracturing".

242- delete "of"

This has been now done.

270/1- reword

This sentence has been now rephrased.

290-thermal shock condition-suggest rethink using this terminology

We agree. "thermal shock" has been now replaced with "intense thermal microfracturing".

301-refs 66,67 refer to mechanically stress induced fractures and faulting, and not thermally induced fracture, so these references should be deleted.

This has been now done.

308--reword to "concluding"

This has been now done.

309/310- reword from "highly crystalline"

This has been now done.

Was any quantitative work done with the observational studies beyond that presented?

The work is predicated on 4 tests, while this represents a lot of information on these tests, they are 4 different tests. how are generalities derived?

Both these questions have been now addressed in the "Concluding remarks" section following the indications of the Reviewer#2.

There should be more to the "conclusions" section, please expound a bit.
This has been now done.

Appendix E

Associate Editor Comments to Author (Professor Axel Schmitt):

The revised version has now accommodated many of the reviewer's suggestions, and in some instances where the authors did not follow the reviewer's criticisms, this was justified for the sake of brevity and focus. Thus overall the manuscript now succeeds in adequately summarizing and discussing a set of interesting experiments and their potential implications for Rn monitoring. However, there are still many shortcomings in the presentation that either are due to deficiencies in the English, or the frequent use of jargon, often obscuring the meaning of the sentences. I have made significant edits in the attached document to improve the manuscript, but there is clearly a need for another careful iteration by the authors, preferably also involving proof-reading by a native speaker, to correct the English and bring the manuscript to an acceptable standard.

Dear Editor,

We are very grateful for the efforts made to improve the quality of the manuscript and for your valuable suggestions. We have now revised the text accordingly. Also, the English text has been reviewed by Dr. John Caulfield, who is a geochemist employed at the School of Earth and Environmental Sciences of the University of Queensland (Brisbane, Australia). The name of John has been also added to the "Acknowledgements" section.

Many thanks again for your time and your help.

Sincerely,

Silvio Mollo and co-authors